# Introgression shapes fruit color convergence in invasive Galápagos tomato

**Matthew JS Gibson[1]\*, María de Lourdes Torres[2,3], Yaniv Brandvain[4], Leonie C Moyle[1]**

[1]Department of Biology, Indiana University, Bloomington, United States; [2]Universidad San Francisco de Quito (USFQ). Colegio de Ciencias Biológicas y Ambientales, Laboratorio de Biotecnología Vegetal. Campus Cumbayá, Quito, Ecuador; [3]Galapagos Science Center, Universidad San Francisco de Quito and University of North Carolina at Chapel Hill, Galapagos, Ecuador; [4]Department of Plant Biology, University of Minnesota-Twin Cities, St. Paul, United States

**Abstract** Invasive species represent one of the foremost risks to global biodiversity. Here, we use population genomics to evaluate the history and consequences of an invasion of wild tomato—*Solanum pimpinellifolium*—onto the Galápagos Islands from continental South America. Using >300 archipelago and mainland collections, we infer this invasion was recent and largely the result of a single event from central Ecuador. Patterns of ancestry within the genomes of invasive plants also reveal post-colonization hybridization and introgression between *S. pimpinellifolium* and the closely related Galápagos endemic *Solanum cheesmaniae*. Of admixed invasive individuals, those that carry endemic alleles at one of two different carotenoid biosynthesis loci also have orange fruits—characteristic of the endemic species—instead of typical red *S. pimpinellifolium* fruits. We infer that introgression of two independent fruit color loci explains this observed trait convergence, suggesting that selection has favored repeated transitions of red to orange fruits on the Galápagos.

\*For correspondence:
gibsomat@indiana.edu

**Competing interests:** The authors declare that no competing interests exist.

## Introduction

The success of colonizing species depends on complex interactions between local environments and the availability of relevant genetic variation. Introduction events are often associated with strong genetic bottlenecks (*Kolbe et al., 2004*; *Colautti et al., 2005*; *Golani et al., 2007*) and reduced effective population sizes, features which may constrain the ability of colonizers to adapt to novel environments and compete with native biota (*Lande, 1988*; *Lee, 2002*). This suggests that biological invasions should rarely follow from introductions (*Queller, 2000*; *Kolbe et al., 2004*), yet successful invasions are nonetheless pervasive (*Kolbe et al., 2004*; *Allendorf and Lundquist, 2003*; *Estoup et al., 2016*; *Comeault et al., 2020*).

Several factors could be involved in this success. Despite intense bottlenecks, diversity could be maintained by other means, including multiple independent introductions (*Facon et al., 2008*; *Kolbe et al., 2004*) or via hybridization with congenerics present in the new habitat (*Ellstrand and Schierenbeck, 2000*; *Lavergne and Molofsky, 2007*; *Reatini and Vision, 2020*; *Stepien et al., 2005*). Of these mechanisms, hybridization might be particularly important for facilitating invasion into island habitats. Hybridization among native and introduced taxa is common on islands (*Carlquist, 1974*), potentially because of limitations on geographic extent, the abundance of generalist pollinators (*Olesen et al., 2002*), and/or frequent anthropogenic disturbance (*Bertolo et al., 2012*; *Lin et al., 2013*; *Long et al., 2014*). In addition, while the geographic isolation of insular habitats

makes them hot spots for species endemism, only a small subset of continental taxa are successful in colonizing remote islands. The resulting incomplete trophic networks provide abundant ecological opportunities for invaders, including reproductive interactions between closely related species. Given these potentially complex contributing factors, describing the occurrence and consequences of invasion is critical for understanding both the dynamics of colonizing populations and for predicting conservation outcomes.

In this study, we investigate the contributions of demographic bottlenecks, single versus multiple introductions, and post-invasion hybridization, to patterns of genomic variation in populations of invasive and endemic tomato species on the Galápagos Islands. Two yellow/orange-fruited tomato species are considered endemic to the islands: *Solanum cheesmaniae* (L. Riley) Fosberg [CHS] and *Solanum galapagense* S.C. Darwin and Peralta [GAL] (*Appendix 1, section S1*). Two red-fruited invasive species from continental Ecuador and Peru are now also documented on the archipelago: *Solanum pimpinellifolium* L. [PIM] and *Solanum lycopersicum* L. [LYC]—the domesticated tomato. Domesticated LYC was almost certainly introduced for agriculture (*Rick, 1963*). Wild species PIM was also likely introduced by early human colonizers (*Darwin et al., 2003*; *Darwin, 2009*), however the timing and source of this introduction is not known. Recent field surveys indicate substantially increased abundance of the invasive species while abundance of the two endemic species has markedly declined over the past two decades (*Darwin, 2009*; *Nuez et al., 2004*; *Gibson et al., 2020*), suggesting that recent demographic shifts may pose an extinction threat to the endemic species. Several factors also indicate a high potential for hybridization between native and invasive species, including overlapping habitats (*Darwin, 2009*; *Nuez et al., 2004*), similar flower morphologies (*Darwin et al., 2003*; *Darwin, 2009*; *Rick et al., 1977*; *Vosters et al., 2014*), and shared pollinators (*Darwin, 2009*). Moreover, all four species are closely related—having diverged less than 500 kya (*Pease et al., 2016*)—and all can be crossed to produce hybrids in the greenhouse (*Rick, 1956*; *Rick and Bowman, 1961*; *Rick and Fobes, 1975*).

Using genomic sequencing data from 174 plants (representing all four species) from the largest islands of San Cristobal, Santa Cruz, and Isabela, and a panel of 132 mainland PIM accessions from across the entire native range on continental South America, we (i) infer the timing, source, and number of invasions by PIM onto the Galápagos and (ii) evaluate evidence for post-colonization gene flow between the four tomato taxa, and its evolutionary consequences. We find that the majority of PIM originated in central Ecuador, are the product of a recent invasion, and are actively hybridizing with an endemic relative. By characterizing fine-scale local ancestry, we find that the emergence of novel orange-fruited plants—which resemble the endemic species in color—in two invasive populations can be explained by endemic introgression at distinct carotenoid loci with known phenotypic effects specifically on fruit color. Our findings reconstruct a recent path of invasion via Ecuador, provide evidence for ongoing interspecific gene flow, and suggest a history of natural selection favoring orange fruits in the island habitat.

## Results

### Sequencing and collections

Sequence data were drawn from 306 individual samples. We performed double-digest RAD (ddRAD) sequencing (using *PstI* and *EcoRI* enzymes) of 174 wild collected individuals from 13 populations of endemic and invasive tomatoes from three islands in the Galápagos archipelago: San Cristobal, Santa Cruz, and Isabela (*Figure 1*; *Table 1*; *Figure 1—figure supplement 1* and *Supplementary file 1a*). We complemented these data with ddRAD reads from 132 mainland PIM (*Figure 2—figure supplement 1* and *Supplementary file 1b*), previously sequenced in *Gibson and Moyle, 2020* using the same enzymes. We recovered 18,573 high-quality RAD loci, each sequenced to an average of 61.4× (s.d. = 35×) in 80% of all 306 samples (*Supplementary file 1c and 1d*). Average insert size was 192 bp (s.d. = 51.7) after adapter and quality trimming. After filtering for depth (>8 reads), 11,297 SNPs were retained. After filtering for LD ($r^2$ < 0.7), 5767 SNPs were retained. Refer to *Supplementary file 1e* for a summary of each filtering step and the analyses for which each dataset was used.

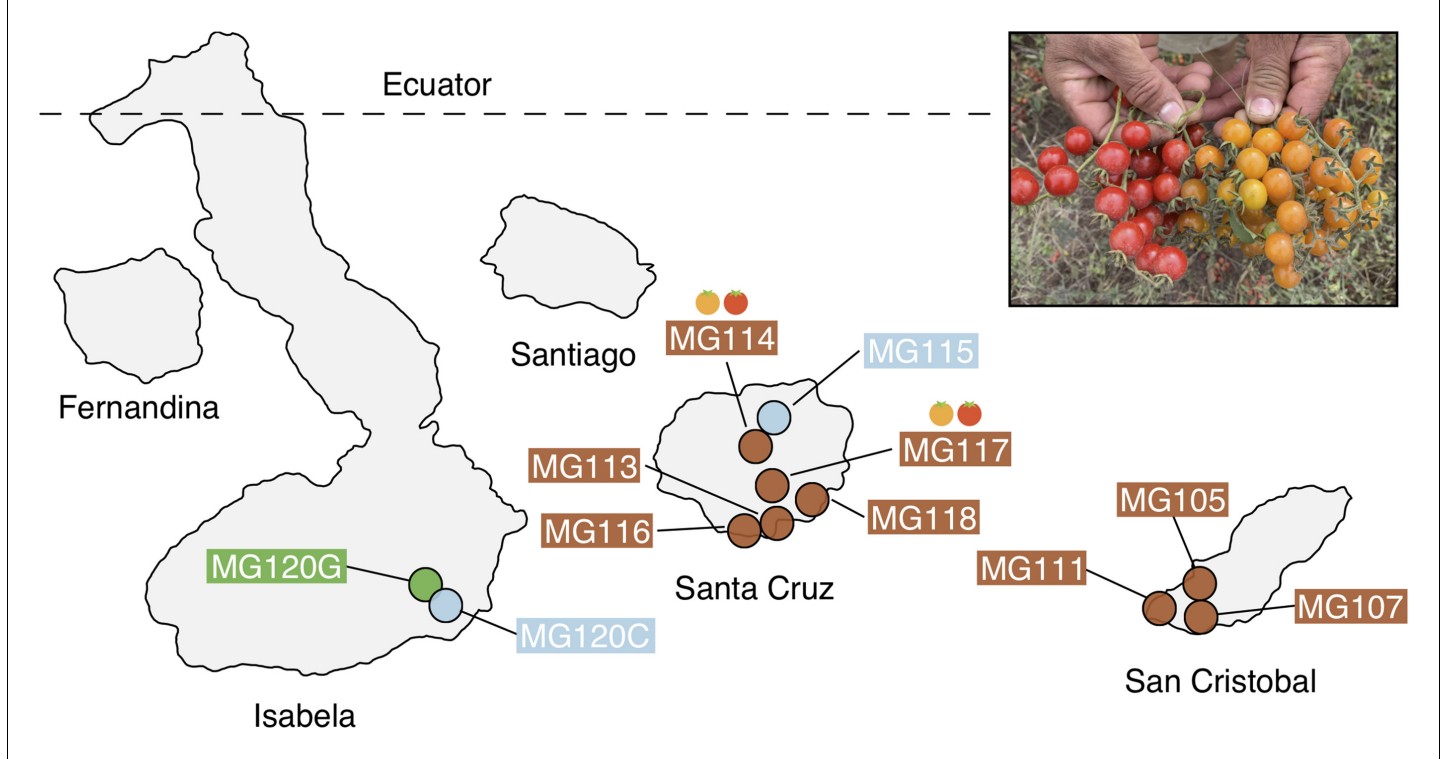

**Figure 1.** Geographic distribution of focal sampling sites on the Galápagos Islands. Inset: Photograph of polymorphic (red/orange) PIM fruits representative of populations MG114 and MG117. For simplicity, LYC populations as well as sampling sites with <8 individuals are not included here. Refer to *Supplementary file 1a* for a full list of collection localities and sample sizes.

The online version of this article includes the following figure supplement(s) for figure 1:

**Figure supplement 1.** Map of all island collection locations.

**Table 1.** Diversity statistics for focal population samples (S = number of segregating sites; $\theta_W$ = Watterson's theta; H = observed heterozygosity; $\pi$ = genome-wide nucleotide diversity).

| Taxa | Population | Island | Endemic | S | $\theta_W$ | H | $\pi$ |
|---|---|---|---|---|---|---|---|
| PIM | Peru | N | N | 32820 | 6302.96 | 0.00025 | 0.00094 |
| | Ecuador | N | N | 21773 | 5544.20 | 0.00033 | 0.00130 |
| | MG105 | Y | N | 1520 | 586.23 | 0.00011 | 0.00009 |
| | MG107 | Y | N | 4434 | 1567.36 | 0.0002 | 0.00034 |
| | MG111 | Y | N | 2562 | 787.93 | 0.00012 | 0.00010 |
| | MG113 | Y | N | 3763 | 1330.17 | 0.00015 | 0.00029 |
| | MG114 | Y | N | 4730 | 1454.69 | 0.00016 | 0.00023 |
| | MG116 | Y | N | 4929 | 1632.189 | 0.00024 | 0.00024 |
| | MG117 | Y | N | 4776 | 1581.52 | 0.00016 | 0.00028 |
| CHS | MG115 | Y | Y | 8540 | 2407.17 | 0.00023 | 0.00045 |
| | MG120C | Y | Y | 854 | 314.22 | 0.0001 | 0.00008 |
| GAL | MG120G | Y | Y | 3282 | 1032.03 | 0.00013 | 0.00014 |
| GAL×CHS | MG120GC | Y | Y | 2857 | 1166.12 | 0.00023 | 0.00026 |
| LYC | MG125 | Y | N | 7219 | 2551.81 | 0.00035 | 0.00063 |
| | MG126 | Y | N | 4567 | 2192.16 | 0.00018 | 0.00052 |

## Genetic data support an Ecuadorian origin for most invasive populations

Using our ddRAD sequencing data for Galápagos and continental PIM, we analyzed population genetic signatures of colonization and characterized the origin and path of invasion into the archipelago. Nucleotide diversity (π in 100 kb overlapping windows; *Figure 2C*) was reduced on average 6.6-fold in island populations relative to mainland accessions (*Table 1*), a pattern consistent with population genetic expectations following colonization.

Genetic variation in the native (mainland) range of PIM is highly geographically structured (*Gibson and Moyle, 2020*; *Figure 2—figure supplement 1*), allowing us to infer a putative origin of PIM lineages invasive on the Galápagos. To do so, we estimated genome-wide patterns of relatedness between invasive and mainland individuals using several methods. A rate-smoothed maximum likelihood tree constructed in *Treemix* (*Pickrell and Pritchard, 2012*) identified Galápagos PIM as monophyletic, and clearly separated island and non-island clades (*Figure 2B*; *Figure 2—figure supplement 2*). In general, pairwise sequence divergence was lower in Galápagos-Ecuador comparisons (average $d_{xy} = 2.3 \times 10^{3}$) than between Galápagos-Peru comparisons (average $d_{xy} = 3.6 \times 10^{3}$;

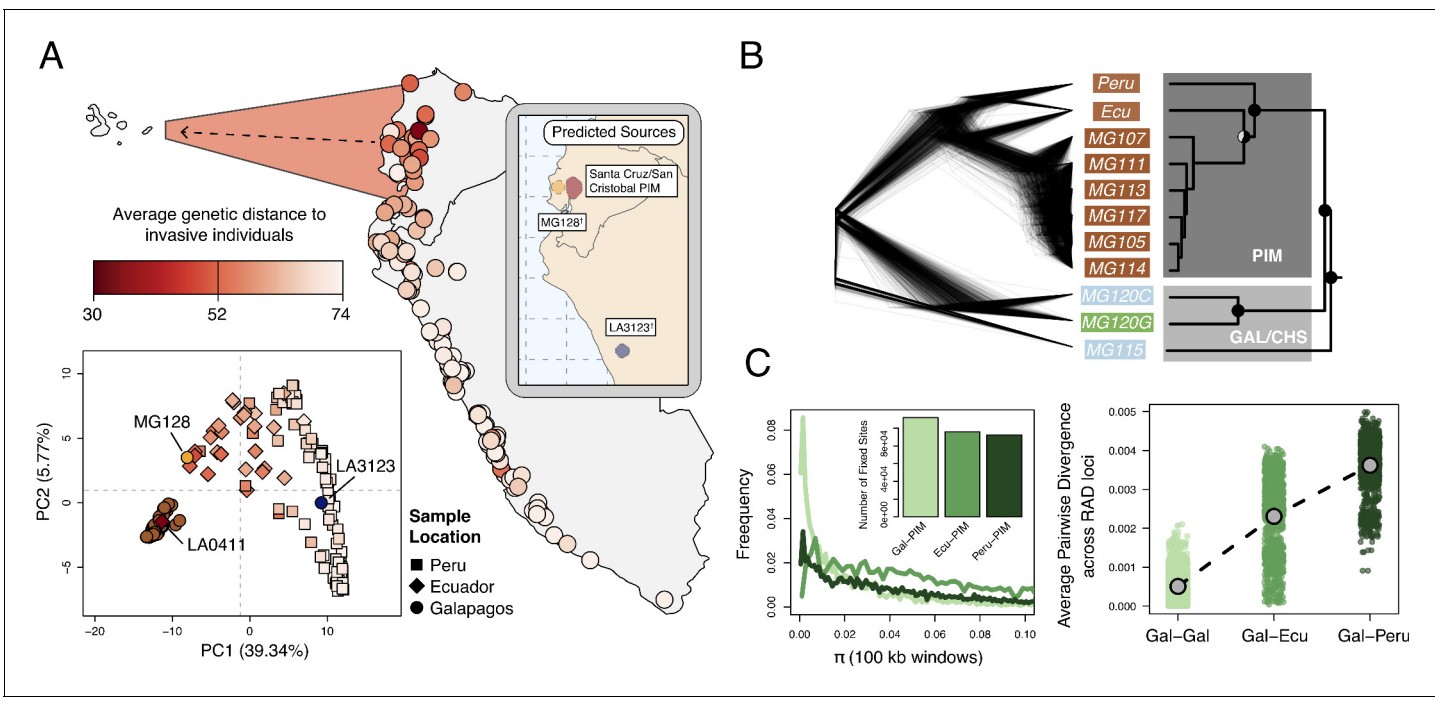

**Figure 2.** Galápagos PIM is the result of a recent invasion from Ecuador. (**A**) *Map*: average genetic distance between Galápagos PIM collections and each of the 132 mainland accessions. *Plot*: multi-locus principal components analysis (PCA). Squares, diamonds, and circles indicate Peruvian, Ecuadorian, and Galápagos collections, respectively. *Inset*: Predicted continental origins for Galápagos PIM collections. Colors are same as shown in the multi-locus PCA ([†]Exact locations vary substantially between runs. Results from a single run are shown). (**B**) Maximum likelihood relationships among focal populations calculated with *Treemix* (allowing no migration). *Left*: inferred trees of 1000 resampled datasets (500 SNPs, with replacement). *Right*: consensus topology. All trees were rate-smoothed (λ = 1). (**C**) Diversity and divergence metrics. *Left*: nucleotide diversity (π) calculated for Galápagos PIM, Ecuador-PIM, and Peru PIM in overlapping 100 kb windows. Invariant windows (π = 0) are truncated and are instead shown in the inset bar plot. *Right*: average pairwise sequence divergence for three PIM comparisons: Gal×Gal, Gal×Ecu, and Gal×Peru. Each point represents a comparison between individuals, averaged over all loci.

The online version of this article includes the following figure supplement(s) for figure 2:

**Figure supplement 1.** Map of mainland collection sites, colored by genetic ancestry cluster as determined in *Gibson and Moyle, 2020*.

**Figure supplement 2.** ML tree of individual samples inferred with RA×ML, using data concatenated across all RAD loci.

**Figure supplement 3.** Six runs of *Locator* (*Battey et al., 2020*) generally support a three-invasion scenario.

**Figure supplement 4.** Pairwise genetic distances at all polymorphic loci.

**Figure supplement 5.** Bootstrapped sampling distributions of Tajima's D for populations MG114 and MG115.

**Figure supplement 6.** Histograms of bootstrapped parameter estimates from dadi for PIM population MG114, using the introgression-masked site frequency spectrum.

*Figure 2C*), and samples showing low genome-wide divergence were clustered in central Ecuador (similar patterns were observed using $F_{ST}$; *Supplementary file 1f*).

To investigate potential source localities for invasive populations at a finer scale, we implemented the software *Locator* (*Battey et al., 2020*) which uses a machine learning algorithm to predict sample origins from genotype data. *Locator* predictions indicated two to three source regions for Galápagos PIM, although the exact locations varied across runs and depended on which island PIM collections were considered (*Figure 2—figure supplement 1*). Santa Cruz and San Cristobal PIM collections were predicted to have originated in central Ecuador; this result was generally consistent across runs, with the consensus being an origin near Los Rios and Guayas provinces in southcentral Ecuador (*Figure 2—figure supplement 3*). Interestingly, we also infer that one mainland accession represents a back migration from the Galápagos to Los Rios (LA0411; *Appendix 1, section S2*), further highlighting the high degree of connectivity between this region and the islands. In contrast, the remaining two samples, LA3123 (a historical collection from Santa Cruz sampled in 1991) and MG128-1 (newly sampled on Isabela), were predicted to have originated in alternative locations, with most runs supporting a Peruvian origin for LA3123 and an Ecuadorian origin for MG128-1 (*Figure 2—figure supplement 3*). The exact origin locations for these samples varied substantially across runs. In general, *Locator* predictions were consistent with the pattern of low pairwise sequence divergence between Galápagos PIM and central Ecuadorian samples, pointing to Ecuador, and perhaps central Ecuador in particular, as the source of the majority of invasive PIM populations on the Galápagos.

Together our data support two to three independent introductions of PIM onto the archipelago, each with variable consequences for current invasive populations: (i) a minor event from Peru [LA3123], (ii) a minor event from Ecuador [MG128-1], and (iii) a major event from central Ecuador that is responsible for nearly all sampled populations.

## Demographic reconstruction supports a recent colonization by PIM on the Galápagos

We used the allele frequency spectrum to model the demographic history of invasive populations. In particular, we evaluated two demographic models using $\delta a \delta i$ (*Gutenkunst et al., 2010*): (i) a neutral model of constant population size and (ii) a two-epoch instantaneous size change model. Since this species is self-fertile (i.e., it lacks genetic self-incompatibility that is present in some wild tomato species; *Rick et al., 1977*), we simultaneously inferred the inbreeding coefficient (F; *Blischak et al., 2020*). The two-epoch model thus included five parameters: epoch one population size ($N_B$), epoch two population size ($N_F$), timing of the first size change ($T_B$), timing of second size change ($T_F$), and the inbreeding coefficient. To limit potential confounding effects due to population structure within PIM, we estimated the folded site frequency spectrum (SFS) of a single population (MG114, which was the most deeply sampled of our PIM populations). We also masked regions of inferred introgression (as detected by our hidden Markov model [HMM], see below) as these can spuriously inflate rare variants and thus bias the inference of a bottleneck and subsequent parameter estimation (*Appendix 1, section S3*). In our masked dataset, we observed a large excess of rare variants (genome-wide Tajima's D = $-0.49 \pm 0.12$; *Figure 2—figure supplement 5*) more consistent with a bottleneck model ($RSS_{Bottle}$ = 0.23; ln(L)=$-18.46$) than a neutral model ($RSS_{Neutral}$ = 16.61; ln(L) =$-26.63$).

We used the best-fit bottleneck model to estimate the timing of the introduction by performing a two-step optimization procedure. We inferred a recent bottleneck occurring 201.66 ($T_B + T_F$) generations in the past (*Table 2*). Bootstrapped CIs for $T_B$ and $T_F$ were relatively small (2–493 gen for $T_B$; 30–401 gen for $T_F$), and medians were in close agreement with optimized estimates (*Table 2*). The bootstrapped median estimate for the time to bottleneck was 200.17, which is less than two generations away from the optimized estimate. CIs for population size parameters as estimated by a non-parametric bootstrap were larger (*Figure 2—figure supplement 6*) and median estimates deviated slightly from the optimized estimates (*Table 2*), although in all cases the optimized size estimates fell within the bootstrapped 95% CIs.

For comparison, we also model the history of an endemic CHS population (MG115) using the same framework as above. The two-epoch model again fit the data better (ln(L)=$-44.61$) than a neutral model (ln(L)=$-74.18$). In addition, we infer a large expansion phase occurring between 1114 and 1845 generations ago followed by a very recent and strong contraction (*Supplementary file 1g*).

**Table 2.** Demographic model estimates for PIM population MG114 inferred using BFGS optimization in $\delta a \delta i$.

95% CI values were obtained from 1000 bootstrap replicates of the site frequency spectrum (SFS). Each estimate is shown in rescaled units (rescaled by $N_{Ref}$ for $N_B$ and $N_F$; and by $2N_{Ref}$ for $T_B$ and $T_F$).

| Parameter | Optimum | Bootstrap median | 95% CI |
|---|---|---|---|
| $N_B$ | 408.32 | 551.52 | 56.32–9325.16 |
| $N_F$ | 4041.63 | 2044.06 | 442.09–26614.5 |
| $T_B$ | 96.52 | 77.15 | 2.37–492.68 |
| $T_F$ | 105.14 | 123.02 | 29.95–401.15 |
| F | 0.23 | 0.17 | 0.01–0.41 |

Compared to our estimate of the timing of the bottleneck in PIM, the inferred expansion in CHS is nearly eight times older. Tajima's D was also higher in MG115 (D = −0.18 ± 0.08) compared to invasive PIM MG114, which is more consistent with neutral expectations and a larger fraction of intermediate frequency alleles.

## Admixture analyses support the occurrence of inter- and intraspecific gene flow

The close evolutionary relationship of PIM, CHS, and GAL, their similar floral morphologies, and the presence of only a single major pollinator on the islands (*Xylocarpii darwini*; **McMullen, 1999**), indicate the potential for interspecific gene flow between tomato species may be high. Key morphological observations also suggest that these species may be exchanging genes (**Darwin, 2009**). In particular, we have previously described a novel fruit color polymorphism in two Santa Cruz PIM populations (MG114 and MG117; **Gibson et al., 2020**), where approximately 40% of individuals have orange instead of their ancestrally red fruits. Orange fruits are very rare in mainland PIM (TGRC passport data; http://www.tgrc.ucdavis.edu) but are diagnostic of the two endemic Galápagos species. Accordingly, we used multiple population genomic methods to investigate evidence of hybridization and introgression in the genomes of island plants, paying special attention to patterns of admixture in the polymorphic PIM populations.

We first examined evidence for recent (early generation) hybrids by evaluating genome-wide signatures in *fastStructure* (**Raj et al., 2014**) and *NewHybrids* (**Anderson and Thompson, 2002**). Interestingly, we find no evidence of early generation CHS×PIM hybrids in either of the polymorphic PIM populations MG114 and MG117 (**Figure 3A**). However, these analyses did detect variable levels of CHS×PIM admixture at the nearby site MG115 (**Figure 3A**), a pattern which is also reflected in principal component space (**Figure 3—figure supplement 1**). Using *NewHybrids*, we classified 4/6 of these admixed plants as first- or second-generation hybrids (**Supplementary file 1h, i and j**).

To complement the above analyses, we employed *Treemix* (**Pickrell and Pritchard, 2012**). The most likely topology inferred by *Treemix* implies three separate admixture events between PIM and CHS: two cases of CHS → PIM admixture and one case of PIM → CHS admixture on Santa Cruz (**Figure 3C**; **Figure 3—figure supplement 2**; **Supplementary file 1k**). This analysis therefore indicates repeated gene flow between PIM and CHS, although how many distinct events were involved is difficult to infer given the high genomic similarity of island PIM populations and recency of the invasion. As independent support for a history of gene flow, we calculated the four taxa D-statistic of **Durand et al., 2011** using *Solanum pennellii* (LA3778) as an outgroup and treating PIM population MG114 and CHS (MG115) as P2 and P3, respectively. We found that D was 0.818 (s.d. = 0.028; bootstrapped p<0.02), indicating an excess of ABBA sites and strong evidence for admixture between island PIM and CHS. D was also significant when other invasive PIM populations—Santa Cruz population MG117 or San Cristobal populations MG107 and MG105—were used as P2, indicating that the detected admixture likely predates the dispersal and differentiation between Santa Cruz and San Cristobal invasive PIM. This is consistent with inferences in the *Treemix* graph, in which admixture events between PIM and CHS involve internal branches that subtend current San Cristobal and Santa Cruz PIM populations (**Figure 3C**).

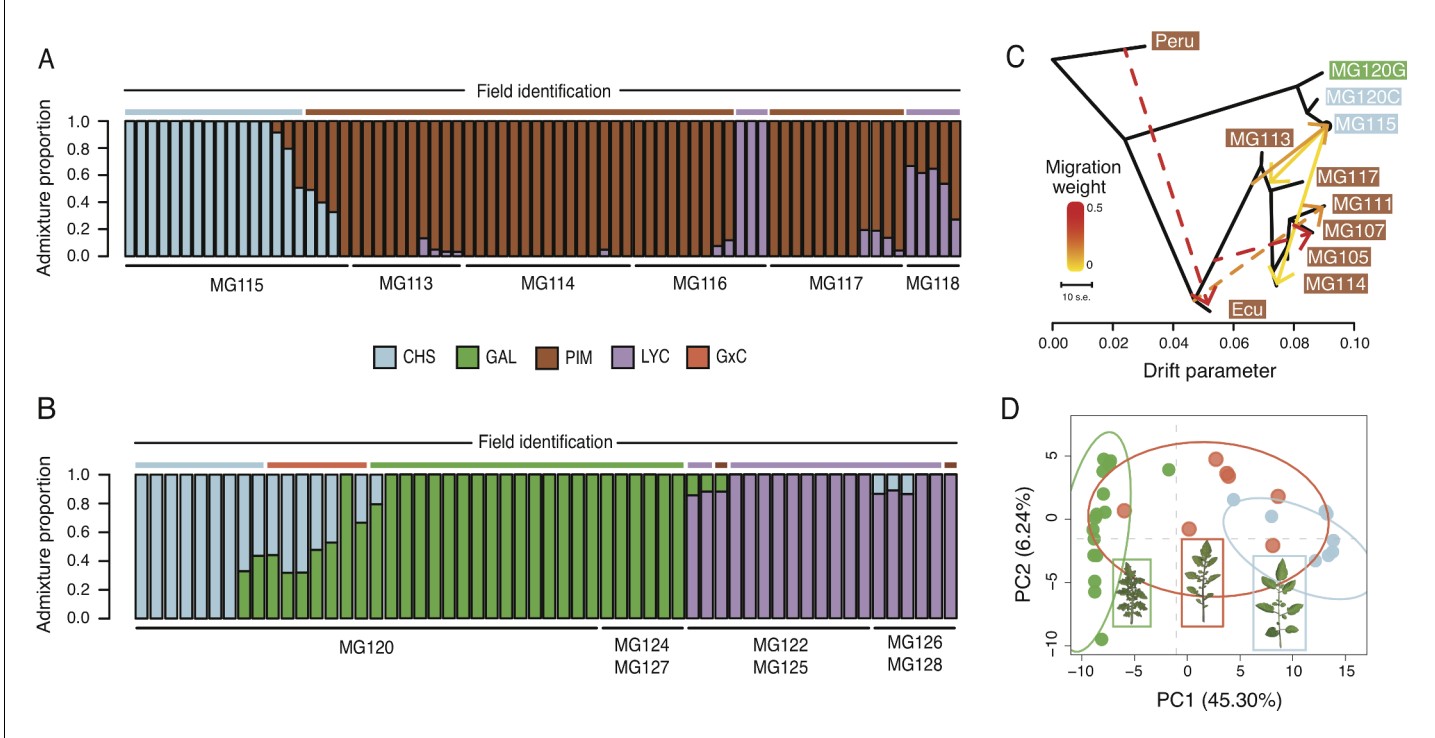

**Figure 3.** Patterns of population genetic structure and admixture on Santa Cruz and Isabela. (A) *fastStructure* inference for all Santa Cruz samples (N = 74). K = 3. (B) fastStructure inference for all Isabela samples (N = 57). K = 3. (C) *Treemix* analysis summary (m = 6; ln[L]=395.08). Solid lines indicate interspecific events and dashed lines indicate intraspecific events. (D) Principal components analysis for samples at site MG120, a hybrid zone between CHS and GAL.

The online version of this article includes the following figure supplement(s) for figure 3:

**Figure supplement 1.** Multi-locus principal components analysis (PCA) for Santa Cruz collections.

**Figure supplement 2.** *Treemix* summary figures for all tested values for m (migration events: 1–8).

Interestingly, *fastStructure* and *Treemix* produced conflicting results regarding evidence for admixture in the polymorphic PIM populations; *Treemix* appears to support this while *fastStructure* does not. To evaluate whether this was due to differences in the detection of more subtle—and potentially older—signals of introgression, we implemented a local ancestry assignment algorithm using a Hidden Markov Model (HMM) to probe for evidence of introgression at a finer scale. Doing so, we found evidence for bidirectional gene flow between CHS and PIM (*Figure 4*; *Figure 4—figure supplement 1*), with inferred introgression being more common in the CHS → PIM direction and in PIM populations that were polymorphic for fruit color. Our HMM detected clear evidence for CHS ancestry within the polymorphic PIM populations MG114 and MG117, reflecting admixture from CHS → PIM. In contrast, inferred admixture in nonpolymorphic PIM (e.g., MG116) was much more restricted. For MG114, CHS ancestry blocks were large (average = 16,235 kb), of varying size (sd = 139.43 kb), and composed on average 3.64% of the genomes of any given MG114 plant (*Figure 4C,D*). Shared ancestry between MG114 and MG115 was dominated by two large CHS haplotypes segregating at moderate to high frequencies on chromosome three (40%; mean size = 51.35 Mb) and chromosome six (20%; mean size = 35.3 Mb; *Figure 4B*). The genomic distribution of CHS ancestry blocks in different individuals indicates they are not independent. For example, on chromosome 3 all but one individual (5/6) carrying the CHS haplotype had identical breakpoints, consistent with them being derived from the same hybridization (and subsequent recombination) event and/or the individuals being closely related (*Supplementary file 1l*). In MG117, CHS ancestry made up 4.36% (sd = 2.56%) of any given MG117 genome and average block size was 9,687.5 kb (*Figure 4C, D*; *Supplementary file 1m*). As with MG114, a large CHS haplotype on chromosome six occurs at a frequency of 0.42. This block varied substantially in size in each individual, and all were discontinuous across the chromosome (i.e., there is an implied double crossover event). Further, two individuals

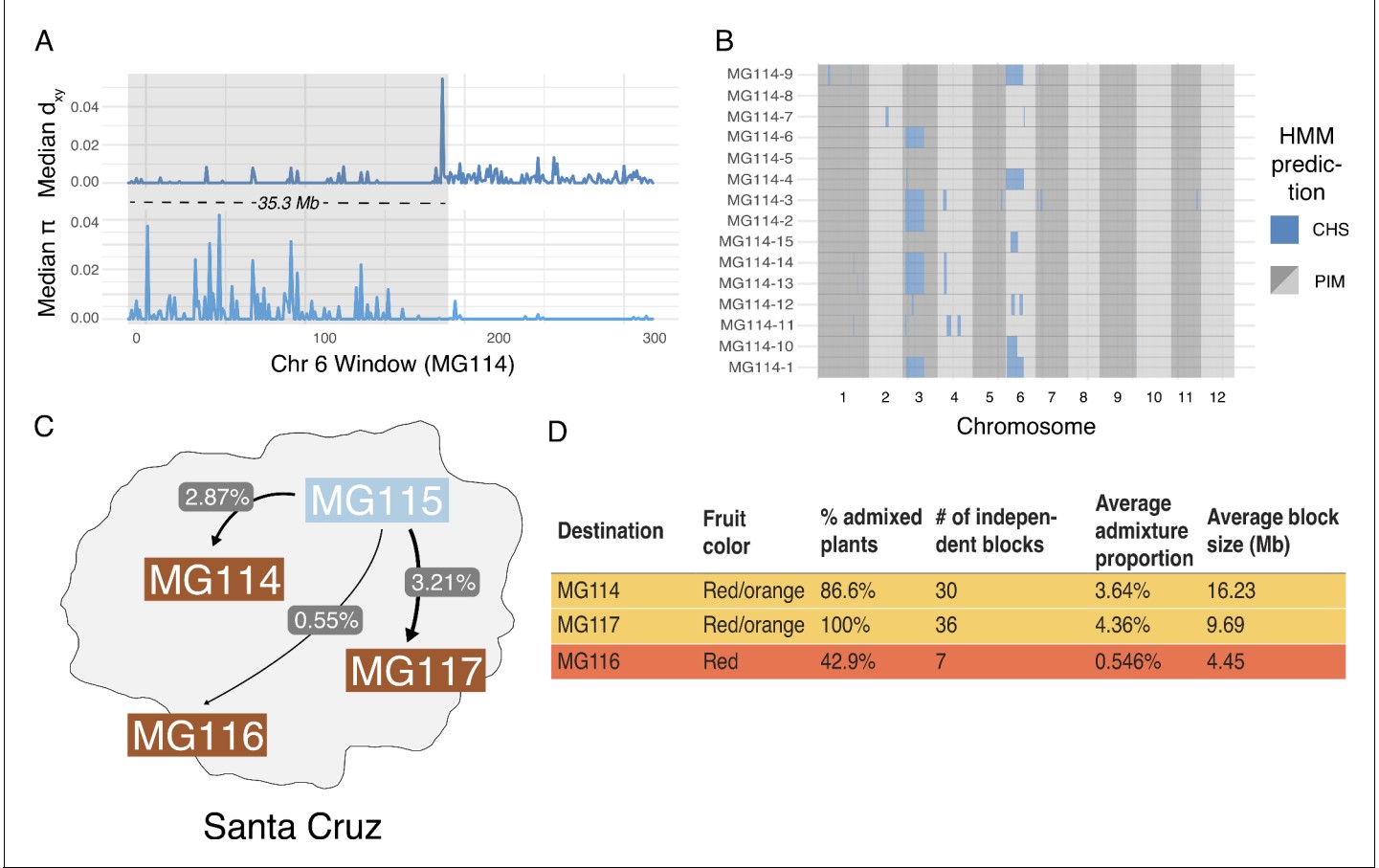

**Figure 4.** Local ancestry assignment using hidden Markov model (HMM) characterizes a history of endemic × invasive introgression. (**A**) Patterns of diversity and divergence along chromosome 6 for an MG114 individual. The region of recent coalescence (low divergence; high diversity) with CHS is annotated in gray. This 20.2 kb block segregates at 20% in MG114. (**B**) Genome-wide HMM predictions for all individuals in MG114. The x-axis is ordered by chromosome and y-axis is ordered by individual. Two large CHS haplotypes segregate at high frequency on chromosomes 3 (40%) and 6 (20%). (**C**) Visual summary of admixture proportions from CHS into three PIM populations. (**D**) Summary of HMM assignment for each PIM population. Populations displaying variation in fruit color (MG114 and MG117) have more CHS ancestry than those which are fixed for the ancestral red state (MG116).

The online version of this article includes the following figure supplement(s) for figure 4:

**Figure supplement 1.** Local ancestry assignment throughout the genomes of CHS plants from population MG115 (Santa Cruz), using MG114 as the PIM reference population.

**Figure supplement 2.** Genome-wide local ancestry in population MG117 as inferred by the hidden Markov model (HMM).

**Figure supplement 3.** Diversity and divergence across the genomes of MG114 individuals (Chromosome 1).

**Figure supplement 4.** Diversity and divergence across the genomes of MG114 individuals (Chromosome 2).

**Figure supplement 5.** Diversity and divergence across the genomes of MG114 individuals (Chromosome 3).

**Figure supplement 6.** Diversity and divergence across the genomes of MG114 individuals (Chromosome 4).

**Figure supplement 7.** Diversity and divergence across the genomes of MG114 individuals (Chromosome 5).

**Figure supplement 8.** Diversity and divergence across the genomes of MG114 individuals (Chromosome 6).

**Figure supplement 9.** Diversity and divergence across the genomes of MG114 individuals (Chromosome 7).

**Figure supplement 10.** Diversity and divergence across the genomes of MG114 individuals (Chromosome 8).

**Figure supplement 11.** Diversity and divergence across the genomes of MG114 individuals (Chromosome 9).

**Figure supplement 12.** Diversity and divergence across the genomes of MG114 individuals (Chromosome 10).

**Figure supplement 13.** Diversity and divergence across the genomes of MG114 individuals (Chromosome 11).

**Figure supplement 14.** Diversity and divergence across the genomes of MG114 individuals (Chromosome 12).

were heterozygous for ancestry at the downstream portion of the haplotype (*Figure 4—figure supplement 2*, *Figure 5—figure supplement 1*, *Figure 5—figure supplement 2*). In comparison to MG114 and MG117, MG116 showed little evidence for shared ancestry. While 3/7 individuals had inferred signals of CHS ancestry, these blocks were generally small compared to MG114 and MG117 (average = 4,450 kb; *Figure 5*) and made up a substantially smaller fraction of the total genome (average admixture proportion = 0.55%; *Figure 4D*). No large CHS haplotypes were segregating in MG116, unlike those observed in MG114 and MG117 (*Supplementary file 1n*).

The size of detected ancestry blocks contains information regarding the timing of gene flow, because it depends on the number of recombination events (generations) that have occurred since an initial hybridization event. We can broadly estimate the age of these haplotypes using a simple logarithmic relationship (*Appendix 1, section S4*; *Lynch and Walsh, 1998*). In MG114 and MG117, age estimates are within the range of 4–12 generations (e.g., the large chromosome 3 and 6 CHS haplotypes in MG114 are estimated at 4.23 and 4.74 generations, respectively). In addition to placing boundaries on when the initial hybridization took place in the past (>4 and up to 12 generations), there is close agreement between age estimates in MG114 and MG117, suggesting that these instances of CHS introgression might have been derived from the same admixture event (*Appendix 1, section S4*).

Relative to the patterns of gene flow from CHS into polymorphic PIM described above, gene flow in the opposite direction (PIM → CHS) was more restricted. The average proportion of PIM ancestry within CHS individuals (at MG115) was 1.62% (s.d. = 2.72%). These results point to a potential bias in the direction of gene flow, with more exchange occurring from CHS into PIM than from PIM into CHS.

In addition to inferred introgression between PIM and CHS, we also found evidence for hybridization/introgression involving the other two taxa: GAL and LYC. In particular, we uncovered a recent history of hybridization between CHS and GAL on Isabela—at *la Laguna de Manzanilla* (*Figure 3B and D*). These two species have been reported as co-occurring at this site since 2000, and hybridization has previously been hypothesized based on allozyme and morphological analyses (*Darwin, 2009*; *Gibson et al., 2020*; *Figure 3D*; *inset images*). Our *fastStructure* and principal components analyses (PCA) of individuals at this site clearly identify nine samples as admixed (*Figure 3B*), mostly corresponding with morphological classifications of intermediacy (*Figure 3D*). Of the nine admixed plants, four appear to be first generation backcrosses, whereas four are F$_2$ (*Supplementary file 1i*). We also identified putative cases of CHS/GAL×LYC admixture in two populations on Isabela (*Figure 3B*), although some of these signals might be the product of unmodeled genetic substructure (*Appendix 1, section S5*). On Santa Cruz, low levels of LYC (domesticated tomato) ancestry were also detected within some PIM populations, including a potential hybrid PIM×LYC population MG118 that was predicted to be entirely first-generation hybrids.

## An introgressed origin for orange fruits in PIM

Because introgression from CHS into PIM was most evident in PIM populations that were polymorphic for fruit color (MG114 and MG117), we took an admixture mapping approach to investigate whether introgression influences fruit color variation in these populations. Specifically, we examined the association between local ancestry across the genome and observed fruit color phenotypes, paying special attention to genomic locations of eight known genes involved in carotenoid biosynthesis (*Supplementary file 1o and 1p*; *Paran and van der Knaap, 2007*). For MG117, we found that the presence of orange fruits correlated perfectly with CHS ancestry at one carotenoid synthesis gene: *CYC-B* on chromosome 6 (*Figure 5B*; *Supplementary file 1o*). This association is significant based on a $\chi^2$ test of independence ($\chi^2$ = 8.33; df = 1; p=0.0039). *CYC-B* is a lycopene beta cyclase and the specific locus known to underlie the lighter (orange) colored fruits observed in the endemic species (*Stommel and Haynes, 1994*).

In contrast, the similarly sized chromosome six haplotype in MG114 (*Figure 4B*) does not include the CHS allele at the *CYC-B* locus. Instead, in MG114 we find that the presence of orange fruits was solely predicted by CHS ancestry at *PSY1* on chromosome 3, the first enzyme in the carotenoid fruit color pathway (*Supplementary file 1p*; $\chi^2$ = 11.12; df = 1; p=0.0009). Although the role of *PSY1* in the coloration of endemic fruits has not yet been studied, loss of function mutations in *PSY1* have been described in LYC and these produce orange fruit color (*Fray and Grierson, 1993*). To further investigate the association of *PSY1* with endemic fruit pigmentation, we used previously published

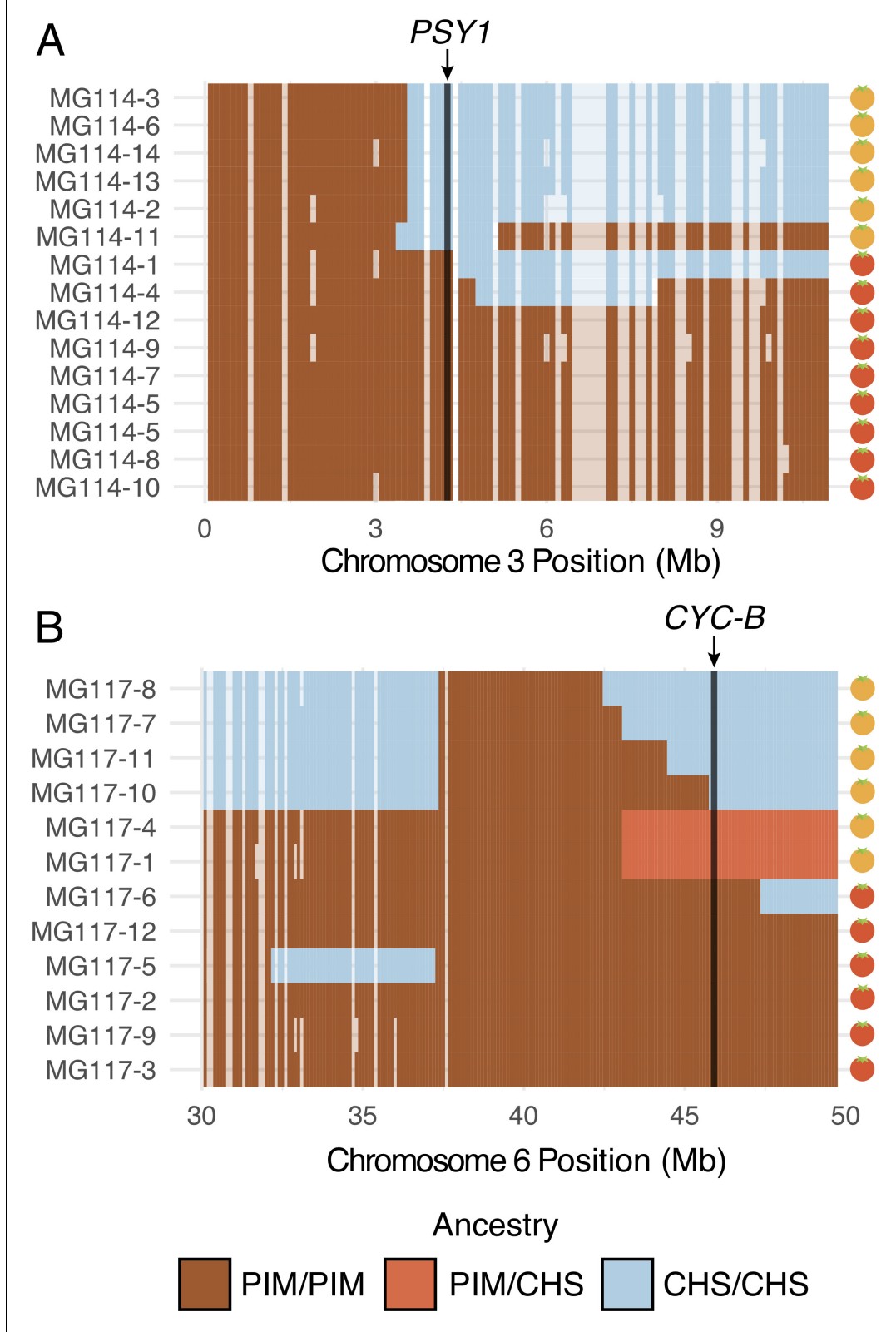

**Figure 5.** Patterns of local ancestry across focal chromosome regions of MG114 and MG117, enlarged to show variation in introgression block break points at color pathway genes. (**A**) CHS ancestry at carotenoid biosynthesis gene *PSY1* on chromosome 3 correlates with observed fruit color variation in MG114. (**B**) CHS ancestry at carotenoid biosynthesis gene *CYC-B* on chromosome 6 correlates with fruit color variation in MG117. Each cell

*Figure 5 continued on next page*

*Figure 5 continued*

represents 100 kb. Empty cells indicate windows with no sequence data. Empty cells are ghost shaded with each ancestry color based on neighboring assignments.

The online version of this article includes the following figure supplement(s) for figure 5:

**Figure supplement 1.** Heterozygosity along chromosome 6 of population MG117.
**Figure supplement 2.** Median divergence estimates between MG117 individuals and MG115 at CYC-B.
**Figure supplement 3.** Coding sequence alignment of *PSY1* for nine wild tomato species (12 accessions).

RNAseq data (*Pease et al., 2016*) to examine variation in *PSY1* across the entire wild tomato clade. *PSY1* is expressed at detectable levels in both endemic species and has a highly conserved coding sequence (1203/1239 shared sites), whose exceptions include a single non-synonymous substitution (62R → W) unique to the endemic clade (*Figure 5—figure supplement 3*). This substitution lies outside of the trans-isoprenyl diphosphate synthase protein domain associated with the enzyme's function, but within the transit peptide signal sequence (residues 1–70), although the specific functional importance of this variant remains to be assessed. Regardless, based on our observed associations between carotenoid biosynthesis loci and fruit color variation, our data support two separate mechanisms underlying the emergence of orange fruits in Galápagos PIM, both of which were likely derived via introgression from CHS.

## Discussion

Biological invasions are one of the foremost threats to global biodiversity, yet we still have a poor understanding of the processes that contribute to invasive colonization success. Here, we studied patterns of genome-wide ancestry and relatedness between endemic, invasive, and continental wild tomato populations in order to reconstruct the history and consequences of a recent biological invasion on the Galápagos. Invasive populations of *S. pimpinellifolium* (PIM) had low levels of genetic diversity and an excess of rare alleles, and we inferred two to three recent introductions onto the archipelago from Ecuador and Peru. As a consequence of this invasion, we uncovered evidence for recent and ongoing gene flow between PIM and the congeneric endemic species *S. cheesmaniae* (CHS). Local ancestry at two key carotenoid loci further supported an introgressed (CHS) origin for orange fruits in at least two invasive populations. Together, our results reconstruct the history of invasion, and infer that the source of convergent phenotypic evolution in the invasive populations is introgression of important functional alleles from endemic relatives.

### Genomic data reconstruct the demographic history of invasion onto the Galápagos Islands

Our analyses identify three independent introduction events, yet only a single event from Ecuador comprises 98% of sampled invasives on the archipelago. The other two introductions—each represented by single plant collections—either did not produce large invasive populations or they result from much more recent introductions which have not yet established broadly. Indeed, our PCA (*Figure 2A*) supports the idea that they are the product of more recent introduction events, as these two collections are more closely related to the mainland PIM populations than plants derived from the primary introduction.

For the primary introduction, although we observed variance in source region predictions (*Figure 2—figure supplement 3*), the consensus prediction supports a southcentral origin near Guayas or Los Rios provinces. Intriguingly, historical data on human migration and trade on the islands also point to this as a likely region for the source of invasive PIM. First, although Ecuadorian colonizers of the Galápagos originate from across the country, one of the earliest and largest bursts of migration coincided with the Tungurahua province earthquake in 1949 (*Toral-Granda et al., 2017*). This province is geographically central and close to Los Rios. Second, the vast majority of all trade between Galápagos and the continent occurred—and continues to occur—from Guayaquil, the second largest city in Ecuador (*Lundh, 2004*; *Toral-Granda et al., 2017*). The surrounding agricultural regions, which include Los Rios, would be the most proximate sources for raw product shipments to the

islands. This historical context provides additional support for our genetic inference of a majority southcentral origin for invasive populations.

Our inferences also clearly implicate humans as the source of PIM introduction. Our demographic reconstruction points to a recent bottleneck and expansion of PIM on the archipelago (*Table 2*), much more recent than our estimate for CHS. Similarly, our inference that LA0411 (a mainland Ecuador accession) is the product of back migration from the Galápagos underscores the recent and likely substantial human influence on the movement of PIM. We conclude that PIM is most likely the result of a recent, human-mediated expansion on the archipelago. Human introduced species represent upward of 70% of all alien plant species on the Galápagos (*Quiroga, 2018*), and PIM has similarly been hypothesized to be the product of a human introduction; however the timing and mode of its introduction—including the role of humans—was not previously known.

## Hybridization as a consequence of invasion onto the Galápagos

One key evolutionary consequence of PIM's introduction onto the Galápagos that emerges from our analyses is its hybridization with endemic congeneric species—primarily CHS. Hybridization has been hypothesized as a mechanism for promoting invasive colonization success, as it could help overcome the adaptive limits that might otherwise be imposed by genetic bottlenecks during the colonization process (the so-called 'genetic paradox' of invasion; *Allendorf and Lundquist, 2003*; *Sakai et al., 2001*). These bottlenecks can be especially severe during introductions onto islands (*Kolbe et al., 2004*; *Colautti et al., 2005*; *Golani et al., 2007*). In addition, several factors indicate the high potential for gene flow specifically between the four studied species (CHS, GAL, PIM, LYC), including their very close evolutionary relationships (all are members of the red-fruited Esculentum subclade within the wild tomatoes; *Pease et al., 2016*), and their incomplete reproductive barriers (*Rick, 1956*; *Rick and Bowman, 1961*; *Rick and Fobes, 1975*). Nonetheless, previous analyses based on handfuls of loci provided conflicting evidence for and against the occurrence of gene flow between species presents on the island (*Nuez et al., 2004*; *Darwin, 2009*).

Our data provide clear evidence for recent hybridization and introgression between all four tomato taxa on the archipelago. Although our focus here is primarily on CHS and PIM, we also find evidence for recent hybridization and/or introgression between CHS and PIM (Santa Cruz), PIM and LYC (Santa Cruz), CHS and GAL (Isabela), and, to a lesser extent, CHS and LYC (Isabela). These patterns suggest that hybridization—both with congeneric endemics (CHS and GAL) and invasives (LYC)—could serve as a source of adaptive genetic variation in invasive PIM.

The most prominent signal of gene flow is between PIM and CHS (*Figure 3*; *Figure 4*; *Figure 5*), including clear evidence for both early generation (F1 and F2) hybrid offspring and older introgression 4–12 generations in the past. Our results indicate (i) that CHS ancestry is maintained in some PIM populations beyond initial hybridization and (ii) that gene flow is ongoing.

The potential consequences of secondary genetic contact are numerous (*Wolf et al., 2001*; *Todesco et al., 2016*). While we do not have direct data on relative fitness of hybrids, the persistence of later generation CHS×PIM hybrids indicates they are not immediately selected against. Indeed, the genomes of most admixed PIM (MG114 and MG117) are consistent with a history of secondary contact and gene flow characterized not by strong hybrid incompatibility, but a less restricted exchange of alleles between species. Furthermore, the nonrandom distribution of CHS ancestry throughout admixed PIM suggests that it may be selectively maintained in certain regions of the genome. Instead of observing a heterogeneous set of CHS alleles in the backcrossed genome of PIM, we find that CHS ancestry is enriched on chromosomes 3 and 6, and absent in much of the rest of the genome, in both MG114 and MG117 (*Figure 4B*; *Figure 4—figure supplement 2*). Moreover, our local ancestry predictions provide evidence that these introgressed regions may contain key genes responsible for the emergence of orange fruit color in MG114 and MG117.

## Orange fruit color in island PIM was derived via introgression from CHS

A key finding of our analyses is that introgression is likely the source of phenotypic convergence on orange fruits that is observed in invasive Santa Cruz PIM. Orange/yellow fruit color is diagnostic for the endemic species (CHS is typically pale yellow; GAL is typically orange) but extremely rare in PIM. At the genetic level, convergence could be based on three potential sources of variation: ancestrally segregating variation, introgression, or via a de novo transition. Of these, ancestral variation is the

least likely: The very few described examples of orange fruits among continental PIM are all located in Peru (e.g., *Sifres et al., 2007*), and none have been reported in the inferred geographic region of origin of this invasion (TGRC passport data; http://www.tgrc.ucdavis.edu). One goal here was therefore to distinguish between introgressed and novel mutation as the source of phenotypic convergence. We did so by mapping the landscape of introgression throughout the genomes of invasive PIM plants, and evaluating its association with observed fruit color variation, and with loci known to underlie this trait in *Solanum* (*Paran and van der Knaap, 2007*). With these data we inferred a unique scenario in which phenotypic transitions to orange fruits in two different invasive PIM populations were each derived from introgression at a distinct carotenoid locus: *CYC-B* or *PSY1*.

Our data in conjunction with existing experimental evidence indicate that *CYC-B* is the causative locus for orange fruits in MG117. Interestingly, *CYC-B* mutants were first identified as natural allelic variation in the endemic species CHS (*Rick, 1956*). Introgression of the CHS *beta* allele at *CYC-B* into LYC causes the accumulation of β-carotene in ripening tissues and the production of orange fruits (*Stommel and Haynes, 1994*). Orange fruits segregate as a single dominant gene, and genotypic variation at this locus explains a large fraction of fruit color variation in experimental crosses (*Rick, 1956*; *Stommel and Haynes, 1994*). Our data show a clear association between CHS ancestry and orange fruit color at this locus (*Figure 5*)—including the observation that individuals heterozygous for ancestry display the dominant phenotype—so we infer that introgression of CHS *CYC-B* into PIM has the same large, dominant effect on fruit color in admixed individuals of this wild species.

Unlike *CYC-B*, *PSY1* was first identified in the spontaneous fruit color mutant *yellow-flesh* in LYC (accession LA2997; *Fray and Grierson, 1993*), and its role has not been directly evaluated in CHS or GAL. Recessive *r* mutants at *yellow-flesh* carry a truncated version of *PSY1* that is unable to convert precursor into phytoene. The resulting fruits accumulate almost no carotenoids and the yellow skin pigmentation is driven primarily by the accumulation of the flavonoid chalconaringenin (*Fray and Grierson, 1993*). Using previously published RNA-seq data (*Pease et al., 2016*), we confirmed the expression of *PSY1* in both endemic species and did not detect any truncation or premature stop mutations. Rather, we identified a single non-synonymous substitution (62R→W) within the transit peptide signal domain, found in both endemic species. Disruption of the transit signal sequence may prevent localization to the chloroplast and thus result in a nonfunctional enzyme, although determining the exact role *PSY1* has in endemic—and by extension PIM—fruit coloration would require future functional confirmation. Regardless, *CYC-B* and *PSY1* in invasive orange-fruited PIM are unequivocally derived from CHS, and current functional knowledge of both loci indicate their effects on fruit color could entirely explain observed phenotypic variation in orange-fruited PIM.

Finally, the ubiquitous lighter (orange and yellow) fruits of the two endemic species, the appearance of convergence toward endemic-like fruit colors in invasive PIM, and the likely independent recruitment of endemic fruit color alleles at *PSY1* and *CYC-B* in MG114 and MG117, together suggest intriguing evidence that lighter fruits may have a specific selective advantage on the islands. The potential environmental basis of this selection is unknown, however differences in fruit dispersal—including disperser color preference(s) and/or fruit color apparency—on the islands versus the continental mainland could be a likely mechanism. Alternatively, at least in the case of *PSY1* which likely involves either a full or partial loss of function, orange pigmentation could arise due to relaxed selection, if it is more costly to produce red fruits and they have no specific advantage in island environments. Future field experiments and fitness measurements will help to distinguish among these selective hypotheses.

## Conclusions

Our results reconstruct a complex and recent history of invasion by wild tomato onto the Galápagos Islands, and highlight the potential importance of gene flow during colonization. Our results also add to an emerging phenotypic convergence literature by highlighting how admixture brought on by anthropogenic change can drive convergence over very short time scales. While the adaptive benefit of orange fruits remains to be evaluated, our finding of two separate molecular mechanisms underlying orange coloration each derived from CHS is highly suggestive that lighter fruit pigmentation is favored in the island environment. This study underscores how the long history of research on the Galápagos Islands continues to enrich our understanding of evolutionary processes in the natural world.

## Materials and methods

### Population sampling and genotyping

We sampled leaf tissue from 13 wild populations of invasive and endemic tomato taxa on the three largest islands of the Galápagos archipelago: San Cristobal, Santa Cruz, and Isabela (*Figure 1*; Table S1). Leaf tissue was dried in silica and DNA was extracted using Qiagen Plant Mini Kits (Qiagen, Valencia, CA). Two double-digest restriction site associated DNA sequencing (ddRAD) libraries were prepared using *PstI* and *EcoRI* enzymes by the Indiana University Center for Genomics and Bioinformatics. Libraries were sequenced across two Illumina NextSeq flowcells (150 bp, paired-end, mind-output). Raw reads were filtered for quality, trimmed of adapter sequence and low-quality bases using *fastp* (*Chen et al., 2018*), and demultiplexed by individual using the *process_radtags* program in *Stacks* (version 2; *Catchen et al., 2013*). Reads were mapped to the *S. lycopersicum* reference genome version SL3.0 using BWA (*Li and Durbin, 2009*). Bam files of 132 continental accessions representing the full species range of PIM (*Figure 2—figure supplement 1*; *Supplementary file 1b*) were jointly reanalyzed with the new samples in *Stacks*. Mapped reads were assembled and variants were called with the Stacks *ref_map* pipeline. Genotype calls made with fewer than eight reads were removed and subsequently we retained only sites having data for at least 80% of all 306 individuals. For all analyses except diversity/divergence calculations ($\pi$, Tajima's D, $d_{XY}$, $F_{ST}$) and *Treemix*, we pruned sites in high LD ($r^2 > 0.7$) using *bcftools*. Lastly, we also evaluated our dataset for two possible sources of bias—the potential effects of allele dropout (ADO) on genotype calls (*Cariou et al., 2016*) and mapping bias arising from using a single reference genome—and confirmed that there is little evidence for either source in our dataset (*Appendix 1, section S6*). All scripts are available at https://github.com/gibsonMatt/galtom (*Gibson, 2021b*; copy archived at swh:1:rev: 1647969c397c5b13d15ab9b5d408bbbab2f6b4a8).

### Nucleotide diversity and divergence estimates

Within-population diversity and divergence estimates across the genome were calculated using the *Stacks* program *populations*. Windowed $\pi$ (*Figure 1C*) was extracted from *Stacks* output. For pairwise comparisons between the islands, Ecuador and Peru (right panel *Figure 1C*), we calculate genome-wide pairwise divergence directly from the assembled RAD loci (samples.fa file) using a custom Python script (http://github.com/gibsonmatt/galtom). For each pairwise comparison between samples, we count the total number of sequence differences and total number of sites for which both samples have data. This choice allows us to conveniently model patterns of diversity between diploid samples in our introgression HMM using a binomial. We also calculated the average genetic distance from each accession to all Galápagos PIM across polymorphic sites in the R package *adegenet* (*Jombart and Ahmed, 2011*; *Figure 1A*).

### Phylogenetic reconstruction

We inferred a maximum likelihood tree of population relationships (*Figure 1B*) using *Treemix* with no specified migration. The *Treemix* input file was generated from a VCF using a custom Python script (http://github.com/gibsonmatt/galtom). We expected abundant phylogenetic discordance both within Galápagos PIM (given its recent divergence from Ecuador) and within the red-fruited tomato clade in general (*Pease et al., 2016*). To this end, we generated 1000 replicate datasets by sampling (with replacement) 500 SNPs from the full dataset. These trees were visualized using *densitree* as implemented in the R package *phangorn* (*Schliep, 2011*). Both the set of replicate trees and the consensus tree were rate-smoothed using *r8s* ($\lambda = 1$), using the chronopl function available in the R package *ape* (*Paradis and Schliep, 2019*). In addition to *Treemix*, we also inferred a maximum likelihood phylogeny of individual sample relationships using *RA×ML* (*Figure 2—figure supplement 2*). We subset our dataset to one individual per population for Galápagos collections and to 15–20 samples per geographic region (Peru and Ecuador) for mainland accessions. We ran RA×ML using the GTRCAT approximation of the general time reversible model of substitution allowing for rate heterogeneity. Twenty-five alternative runs from distinct maximum parsimony trees were performed, from which we selected the best single tree.

## Demographic inference

We modeled bottleneck demographic histories from the SFS using $\delta a \delta i$ (*Gutenkunst et al., 2009*), calculating the folded SFS using *easySFS* (https://github.com/isaacovercast/easySFS, copy archived at swh:1:rev:b866269f5813ef7cd8a12c7727048f993da8e9ff; *Gibson, 2021a*), 30–14 chromosomes, striking a balance between the number of segregating sites and levels of missing data in frequency bins. We chose this population because the larger number of samples (2N = 30) relative to other PIM populations afforded more statistical power. Prior to calculating the SFS for MG114, we removed regions inferred as introgressed by our HMM, as we found that a large fraction of singleton and doubleton sites in MG114 were shared with CHS (*Appendix 1, section S3*). These sites would be spuriously interpreted as de novo mutations derived in PIM post-colonization, thereby biasing our parameter estimates. See *Appendix 1, section S3* for further discussion of our filtering scheme and its effects on $\delta a \delta i$ estimates. For population MG115, we down-sampled to 24 chromosomes. For each population, two models were evaluated: (i) a neutral model of no population size change and (ii) an instantaneous bottleneck model. The parameters of the bottleneck model are described in the results. We used the BFGS algorithm and a two-step optimization procedure to explore the demographic parameter space of the PIM bottleneck model. Step i consisted of 200 replicate runs with twofold parameter perturbation. Step ii then consisted of 100 refinement runs of the optimizer with no perturbation and initial parameter values equal to the maximum likelihood estimates from step i. We used multiple diffusion grid sizes in $\delta a \delta i$ (30, 40, and 60) as recommended in the manual to facilitate extrapolation. To estimate confidence intervals for our parameters, we performed a nonparametric bootstrapping procedure (assuming independence among sites) by sampling randomly from the observed SFS 1000 times. Since we filter for LD, we consider this method to be appropriate.

Time in $\delta a \delta i$ is represented in units of $2N_{ref}$ generations. To convert from these coalescent time values to numbers of generations, we estimate $N_{ref}$ as $\theta/4$ $\mu$L, where $\theta$ is the population scaled mutation parameter (estimated by $\delta a \delta i$; 68.40 for MG114), $\mu$ is the per-generation mutation rate (assumed here to be $1 \times 10^{-8}$ muts/bp/generation), and L is the length of queried sequence (5,806,952 bp; estimated from the data as the total number of bases where at least one sample in the focal population had data). Bottleneck and final effective population sizes were similarly converted from relative to absolute values using $N_{ref}$.

## Inferring gene flow between contemporary island populations

We used several independent methods to characterize genetic structure in our dataset. First, we applied *Treemix* (*Pickrell and Pritchard, 2012*) to assess evidence for broad signatures of admixture among populations. Because we were interested in understanding the history of PIM, we subset our full SNP dataset to exclude LYC populations since the exact taxonomic status of these samples is unclear and does not help address questions of gene flow between PIM and endemic species. Furthermore, we remove PIM populations with fewer than eight samples. We ran *Treemix* with several values for m (0–8), the migration parameter which determines how many reticulation branches are allowed. Based on the likelihoods provided by *Treemix*, we determined that a migration parameter of 6 was most appropriate. Increasing m led us to infer more intraspecific migration within PIM, but no additional interspecific events could be inferred and the increase in data likelihood was marginal (*Figure 3—figure supplement 2*; *Supplementary file 1j*).

We also employed model-based (*fastStructure*; *Raj et al., 2014*) and non-model-based (multilocus PCA) methods to evaluate genetic structure. We ran both methods using the LD-filtered dataset of 5767 SNPs and on each island separately. For each island, we ran *fastStructure* for values of k between 1 and 7. We then chose an appropriate model complexity using the *chooseK.py* script supplied with *fastStructure*. After a value for k was chosen, we evaluated the stability of the ancestry assignments across 5–10 separate runs. The multi-locus PCA was implemented in the R package *adegenet*.

For select populations with evidence for admixture, we ran *NewHybrids* (*Anderson and Thompson, 2002*) to identify any early generation hybrid individuals ($F_1$, $F_2$, and backcrosses). *NewHybrids* was run using the same LD-filtered dataset as in *fastStructure* and PCA (script for converting from vcf to *NewHybrids* format is provided at http://github.com/gibsonmatt/galtom). For each *NewHybrids* run, we specify three groups: a parent population A group, a parent population B group, and an admixed group. For the CHS×PIM interaction on Santa Cruz, parent populations were defined as

non-admixed MG115 and MG113 individuals. For the CHS×GAL interaction on Isabela, parent populations were MG120C and MG120G. For each population, we ran the Markov chain for at least 6000 iterations. Individual assignments were not sensitive to the choice of priors. Lastly, four-sample D-statistics (*Durand et al., 2011*) were calculated using the *compD* software package (https://github.com/stevemussmann/Comp-D; *Mussmann, 2020*), using *S. pennellii* (LA3778) as an outgroup species.

## Introgression analysis

We implemented an HMM to identify fine-scale genomic signatures of introgression. Although RAD sequencing is not optimal for genome scanning due to lower marker density, recent signatures of introgression should be large and detectable. Nonetheless, we acknowledge the fact that our sequencing methods may not allow for full characterization of the landscape of introgression. In our analysis we leveraged the fact that, in regions of recent introgression, genetic diversity between samples within the destination population ($\pi$) should be elevated relative to diversity between samples in the source and destination populations ($d_{XY}$). In other words, regions of recent interspecific coalescence should resemble individuals from the introgressing population more than individuals in the destination population. For each pairwise comparison between samples, we thus have an estimate of $\pi$ (if the samples are from the same population) or $d_{XY}$ (if they are from different populations) as well as the raw number of sequence differences and total sites with data.

Our HMM featured three hidden states: (i) CHS ancestry, (ii) PIM ancestry, and (iii) heterozygous CHS/PIM ancestry, for which we used $\pi$, $d_{XY}$, and the mean of $\pi$ and $d_{XY}$ (m) in nonoverlapping 100 kb windows, respectively, to calculate emission probabilities. The HMM was applied to each individual separately, using pairwise sequence comparisons between it and all other focal samples to calculate probabilities. We used three binomial models to obtain these probabilities (see *Appendix 1, section S7* for a detailed description of the model). For each chromosome and for each focal comparison, we found the most likely hidden state path for a given sequence of $\pi$ and $d_{XY}$ using the Viterbi algorithm, controlling for underflow by operating in log-space. Because of the coarse scale (100 kb windows) used in ancestry assignment, our HMM struggled to identify smaller genetic signals potentially consistent with introgression. *Figure 4—figure supplement 3* provides a per-window look at how ancestry assignment correlated with patterns of diversity and divergence in MG114. Patterns of diversity/divergence vary substantially even between adjacent windows and thus it is likely that we are not capturing sub-100kb signals of introgression. Nonetheless, the large size of the blocks we do identify allows us to be confident they are the result of introgression rather than ILS.

## Acknowledgements

We thank the Galápagos Science Center staff on San Cristóbal for logistic and permitting support, and the Galápagos National Park for assistance locating and sampling endemic populations. On-site field support was provided by Marcelo Loyola and Genaro Garcia. We also thank two anonymous reviewers for valuable comments that greatly improved the manuscript. This work was supported by a US National Science Foundation award (IOS 1127059) to LCM and the Indiana University Brackenridge award to MJSG. All field collections were made with appropriate permits and prior authorization by the Galápagos National Park and Ecuadorian Ministry of Environment (Permits PC-40–18 and PC-72–19).

## Additional information

### Funding

| Funder | Grant reference number | Author |
| --- | --- | --- |
| National Science Foundation | IOS 1127059 | Leonie C Moyle |
| Indiana University Bloomington | Brackenridge | Matthew JS Gibson |

The funders had no role in study design, data collection and interpretation, or the decision to submit the work for publication.

## Author contributions

Matthew JS Gibson, Conceptualization, Resources, Data curation, Software, Formal analysis, Supervision, Funding acquisition, Validation, Investigation, Visualization, Methodology, Writing - original draft, Project administration, Writing - review and editing; María de Lourdes Torres, Conceptualization, Supervision, Investigation, Methodology, Writing - review and editing; Yaniv Brandvain, Software, Methodology, Writing - review and editing; Leonie C Moyle, Conceptualization, Funding acquisition, Investigation, Methodology, Writing - original draft, Writing - review and editing

## Author ORCIDs

Matthew JS Gibson  https://orcid.org/0000-0001-7855-1628
María de Lourdes Torres  https://orcid.org/0000-0001-7207-4568
Leonie C Moyle  https://orcid.org/0000-0003-4960-8001

## Decision letter and Author response

Decision letter https://doi.org/10.7554/eLife.64165.sa1
Author response https://doi.org/10.7554/eLife.64165.sa2

# Additional files

## Supplementary files

• Supplementary file 1. Supplementary tables. (**a**) Full list of population collection sites and their geographic coordinates. (**b**) List of mainland collection sites. (**c**) Stacks ref_map assembly summary. (**d**) Stacks per-sample assembly summary. (e) Summary of sequence and genotype filters. (**f**) Mean Fst between focal island populations and mainland groups. (**g**) Demographic model estimates for PIM population MG115. Values inferred using δaδi. 95% CI values were obtained from 1000 bootstrap replicates of the site frequency spectrum (SFS). Each estimate is shown in rescaled units (rescaled by NRef for NB and NF; and by 2NRef for TB and TF). (**h**) Full likelihoods for NewHybrids classifications of CHS×PIM admixture on Santa Cruz. MCMC was run for 6000 steps. The most likely classification for each individual is shown in bold. (**i**) Full likelihoods for NewHybrids classifications of CHS×GAL admixture on Isabela. MCMC was run for 6000 steps. The most likely classification for each individual is shown in bold. (**j**) NewHybrids MCMC summary. Only the most likely genotype category classifications are shown. Refer to (**e**) for alternative class probabilities. (**k**) Likelihoods for different Treemix runs. (**l**) Summary of inferred introgression blocks for population MG114 (polymorphic). Shaded groups indicate blocks that are likely the same age/the result of the same hybridization event based on break points. (**m**) Summary of inferred introgression blocks for population MG117 (polymorphic). Shaded groups indicate blocks that are likely the same age/the result of the same hybridization event based on break points. (**n**) Summary of inferred introgression blocks for population MG116 (fixed, red fruited). (**o**) Population MG117 ancestry at carotenoid biosynthesis loci, as inferred by the hidden Markov model (HMM). The genomic location of each locus was determined based on the *Solanum lycopersicum* reference build SL3.0 and ITAG3.0 annotation. The association between ancestry and fruit color at CYC-B is significant based on a $\chi^2$ test of independence ($\chi^2 = 8.333$; df = 1; p=0.00389). The association at LCY-B is not statistically significant ($\chi^2 = 0.6$; df = 1; p=0.4386). (p) Population MG114 ancestry at carotenoid biosynthesis loci, as inferred by the HMM. The genomic location of each locus was determined based on the *S. lycopersicum* reference build SL3.0 and ITAG3.0 annotation. The association between ancestry and fruit color at PSY1 is significant based on a $\chi^2$ test of independence ($\chi^2 = 11.123$; df = 1; p=0.00085). The association at LCY-B is not statistically significant ($\chi^2 = 2.934$; df = 1; p=0.08673).

• Transparent reporting form

## Data availability

Raw, demultiplexed ddRAD reads have been deposited under NCBI BioProject PRJNA661300 and will be available once processed by NCBI. Genotype files, associated datasets, and analysis scripts have been deposited on Dryad (https://doi.org/10.5061/dryad.2v6wwpzkm). Additionally, data posted to Dryad can also be accessed at https://github.com/gibsonMatt/galtom (copy archived at https://archive.softwareheritage.org/swh:1:rev:1647969c397c5b13d15ab9b5d408bbbab2f6b4a8).

The following dataset was generated:

| Author(s) | Year | Dataset title | Dataset URL | Database and Identifier |
|---|---|---|---|---|
| Gibson MJ, Torres MdL, Brandvain Y, Moyle LC | 2020 | Data from: Reconstructing the history and biological consequences of a plant invasion on the Galápagos islands | https://doi.org/10.5061/dryad.2v6wwpzkm | Dryad Digital Repository, 10.5061/dryad.2v6wwpzkm |

The following previously published datasets were used:

| Author(s) | Year | Dataset title | Dataset URL | Database and Identifier |
|---|---|---|---|---|
| Pease JB, Haak DC, Hahn MW, Moyle LC | 2016 | Phylogenomics reveals three sources of adaptive variation during a rapid radiation | https://doi.org/10.5061/dryad.182dv | Dryad Digital Repository, 10.5061/dryad.182dv |
| Gibson MJS, Moyle LC | 2020 | Regional differences in the abiotic environment contribute to genomic divergence within a wild tomato species | https://doi.org/10.5061/dryad.8gtht76k4 | Dryad Digital Repository, 10.5061/dryad.8gtht76k4 |

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

## Appendix 1

### S1. Field collections and taxonomic treatments
#### S1.1. Collections and DNA extraction

We visited San Cristóbal, Santa Cruz, and Isabela—the three most populated islands in the archipelago. These islands contain 40 documented TGRC locations (TGRC passport data) and these sites, as well as those described in *Darwin, 2009* and *Nuez et al., 2004*, were searched during our expeditions. We looked for plants within a 200 m radius of each previously documented collection site, and we also searched for additional undocumented populations. Populations were sampled across linear transects and detailed photographs of leaf, fruit, and flower morphology were taken of all sampled individuals against color and length standards. The latitude and longitude of each collection location was logged using a smartphone. Each site and the identity of species found within are described in Table S1. For extracting DNA, three to five leaves were sampled from each plant and immediately dried in silica gel. Tissue disruption was performed with mini-pestles and DNA for each sample was extracted using Qiagen Plant Mini kits (Qiagen, Valencia, CA) at the Galápagos Science Center (San Cristobal, Galápagos, Ecuador). Our sequencing pipeline is described in the main text.

#### S1.2. Taxonomy

Population identity was determined based on the taxonomic treatments described in *Darwin et al., 2003*, with the exception of LYC, which we separated into two forms—LYC (domesticated tomato) and LYC var. *cerasiforme* (CER; cherry tomato; *Rick, 1956*)—to maintain consistency with *Nuez et al., 2004*. Ripe fruit color was the primary trait used in species identification, with the exception of individuals designated as orange-fruited *S. pimpinellifolium*, based on all other characters. We additionally measured nine leaf traits—leaf length, leaf width, terminal leaflet length, leaflet count, interjected leaflet count, petiole length, internode length, leaflet length, and leaflet width previously identified as diagnostic by *Darwin et al., 2003*.

### S2. LA0411 and our inference of back migration
#### S2.1. Original collection notes for LA0411

Charles *Rick, 1956*: 'Pichilingue. 200 m. *Lycopersicon pimpinellifolium* or *Lycopersicon esculentum* var. *cerasiforme*? Weedy vigorous form growing as weed in corn field N of station buildings. Population of 15 plants found here; five had ripe fruit, long internodes, leaves more elaborate than pimp. Few long hairs at growing point like Santa Cruz pimpinellifolium. Alternation all = 3 or ±3. Flowers small, stigmas all exserted 1 mm or more. Population seems to be entirely uniform. Inflorescence all first order, racemose. Fruits large for pimp = 1.5 cm, red. No vectors seen. Long tailed thick billed black birds seen near patch. These said to eat wild tomatoes near Guayaquil cement factory. Odor = esculentum. Anthocyanin = dark'.

#### S2.2. Evidence for back migration

Based on patterns of genomic relatedness to Galápagos and surrounding PIM accessions, we infer that mainland Ecuadorian accession LA0411 (collected in 1956 by Charles Rick) is the product of a back migration from the Galápagos to mainland Ecuador. This accession showed a particularly strong resemblance to Galápagos PIM (average $d_{xy}$ = 0.0006) and was also divergent from neighboring mainland Los Rios accessions (*Figure 2—figure supplement 4*). This accession was indeed noted as morphologically similar to Galápagos PIM tomato collections when sampled on mainland Ecuador in 1956 (TGRC passport data; http://tgrc.ucdavis.edu; S2.1, above). This inference has two implications for understanding the history of PIM and other invasive species on the islands. First, it sets an upper bound on the timing of the initial introduction of PIM to the Galápagos as no later than 1956, as it must have already been established there prior to a back migration event. Second, it highlights the substantial connectivity between this region (the putative source of invasive PIM) and the Galápagos in general.

This finding does not affect our analyses of invasive population origins, including our inference that most invasive PIM have an Ecuadorian source. Invasive PIM have high genetic similarity with

many accessions in this region of Ecuador (*Figure 2A*). Furthermore, running *Locator* without accession LA0411 produces results identical to those described in the main text.

## S3. The impact of recent introgression on demographic inferences with $\delta a \delta i$

A recent history of introgression from CHS/GAL into invasive PIM populations could bias $\delta a \delta i$ parameter estimates toward older dates by introducing low-frequency alleles that would incorrectly be interpreted as de novo mutations derived post-expansion. To examine this possibility, we determined whether rare alleles in MG114 were private (i.e., informative of the time since bottleneck) or shared with CHS (e.g., from introgression). We found that a large portion (52%) of singleton and doubleton SNPs in MG114 were shared with CHS (MG115), suggesting that introgressed variants could substantially affect our parameter estimates of the occurrence and timing of demographic changes within invasive PIM. For this reason, prior to model fitting we removed SNPs within all genomic regions where at least one MG114 individual had evidence for CHS ancestry based on our HMM. Any region inferred to be of CHS ancestry in any individual was removed from the dataset for all individuals. This resulted in 365 sites being filtered from the SFS relative to the original dataset.

## S4. On the timing of introgression

The size of introgression blocks contains information about their age. Over time, recombination will break up contiguous blocks into smaller tracts, resulting in a mosaic of ancestry throughout the genome. We can roughly estimate the age of any given block if we assume that an initial hybridization event was followed by subsequent backcrossing to non-admixed parental individuals. Using a simple logarithmic relationship, the expected time t required to arrive at a proportion *p* of the donor genome after repeated backcrossing is estimated as:

$$\log\left(\frac{1}{\mathrm{p}}\right)/\log(2)$$

via *Lynch and Walsh, 1998*. As block size gets smaller, the expected time since hybridization increases.

The distribution of introgression tract sizes and the concordance of break points between blocks within and between populations are indicative of non-independence among blocks. Such a pattern makes any detailed assessment of the timing of these events difficult, and our broad estimate makes several simplifying assumptions. The occurrence of selfing and/or small population sizes may also affect the relationship between block length and age of introgression. Nonetheless, it is clear that many of the observed patterns of CHS ancestry throughout the genomes of MG114 and MG117 plants were derived from very recent (and likely shared) events. We further investigated the degree to which these histories of introgression were shared by comparing the ancestry of each population window-by-window. Specifically, we called windows of shared CHS ancestry in MG114 and MG117 as those where at least one individual in each population was assigned as CHS by our HMM. Across the genome, 250 windows were inferred to be of CHS ancestry in both populations, representing 26.8% of all CHS ancestry in MG114% and 35.4% of all CHS ancestry in MG117. These patterns imply a complex and partially shared history of admixture in MG114 and MG117. This inference is consistent with several of our other analyses, including *Treemix* (*Figure 4*) and D-statistics which also pointed to a shared basis to detected patterns of admixture. At the resolution which we are able to assign local ancestry, our data also firmly indicate that endemic variation is maintained within PIM beyond the first or second generation of hybridization.

## S5. Further details on evidence for gene flow

We use several statistical methods for detecting gene flow between invasive and endemic populations. Our primary focus was on characterizing admixture between CHS and PIM on Santa Cruz (described in text), however several additional patterns are also worth presenting. These are discussed below.

## S5.1. Treemix

In addition to the two cases of interspecific PIM×CHS admixture, three cases of intraspecific admixture within PIM were inferred using *Treemix*: one from Peru PIM into Ecuador PIM, one from the ancestral Galápagos branch to MG107, and one from Ecuador PIM to MG111. It is difficult to interpret the factors responsible for these inferred events, although they may not all reflect true admixture cases. The edge between Ecuador and Peru PIM is most likely a byproduct of our distinction between these groups; in reality the relatedness among mainland samples reflects a pattern of IBD across latitude. The edges leading to MG107 and MG111 could suggest the occurrence of additional minor introduction events nested within the major Ecuador event. Such a scenario might be plausible given the known substantial trade between the Galápagos and central mainland Ecuador, and the occurrence of at least one reintroduction event. However, we have no independent support for such additional events. Increasing m in *Treemix* had the effect of inferring additional intraspecific migration within PIM, but did not infer any additional between species admixture (see *Figure 3—figure supplement 2* for additional *Treemix* run summaries) and the increase in data likelihood was marginal (*Supplementary file 1k*). This indicates that the precise parameter choices in these analyses do not change the number of introgression events inferred between invasive PIM and the endemic island species, beyond those that described in the main text.

## S5.2. *S. lycopersicum*

In addition to MG118 (population of F1 PIM×LYC plants) described in text, other weaker signals of LYC ancestry in PIM (e.g., in MG113, M114, M116, and M117; *Figure 4A*) may also be the result of post-colonization admixture, however they may also represent latent (unmodeled) population structure within PIM and/or reflect the hybrid ancestry of LYC var. *cerasiforme* (*Ranc et al., 2008*). Increasing K from 3 to 4 in *fastStructure* swapped the minor LYC ancestry fractions seen in MG113, MG116, and MG117 for a fourth ancestry class. However, the minor LYC component seen in MG114 as well as the 50/50 LYC×PIM MG118 population retained their original LYC classifications. It is difficult to interpret this given the complex and unresolved history of LYC var. *cerasiforme* (*Ranc et al., 2008*), however it may indicate (i) the presence of two or more distinct LYC lineages on the Galápagos or (ii) a contribution by *S. galapagense* (GAL; not sampled on Santa Cruz but historically present). The former scenario may be expected if multiple varieties of domesticated seed have been used for cultivation on the archipelago.

Separate from the clear case of CHS×GAL admixture on Isabela at *la laguna de manzanilla* (discussed in the main text), additional signals of admixture between CHS/GAL (MG120)×LYC (MG122 and MG126) were also detected (*Figure 4B*). In MG122, increasing K from 3 to 4 clearly separates the three inferred GAL×LYC individuals into a separate non-admixed subcluster, suggesting that the inferred GAL ancestry in these samples likely reflects latent substructure within LYC. In contrast, even when increasing K from 4 to 5, the three inferred CHS×LYC samples in MG126 retain their minority CHS component, consistent with a real signal of shared ancestry between CHS and LYC for these individuals. Whether this is the result of gene flow post-colonization of LYC on the islands or a reflection of breeding history (CHS germplasm has been used as a source for certain beneficial crop traits; *Rick, 1967*; *Stommel and Haynes, 1994*) is unclear.

## S6. Evaluating the potential impact of allele dropout and reference mapping bias

Allele drop out (ADO) is a potential source of bias in restriction enzyme-based genotyping protocols. When polymorphisms occur in restriction cut sites, the RAD tag can 'drop out', potentially masking true SNPs present in linked haplotypes. ADO can convert true heterozygous genotypes into homozygous calls. It has been well documented in simulations and real data (*Cariou et al., 2016*), yet methods for dealing with it are lacking. The general observation is that, when polymorphism is high, the probability of an SNP occurring in a cut site increases, resulting in fewer true heterozygotes being identified and an overall bias toward observing lower levels of diversity.

To assess whether ADO is having a substantial effect in our dataset, we examined the relationship between sequencing depth and observed locus heterozygosity, since loci experiencing dropout are expected to be sequenced to lower depths. We observe no significant association between depth

and heterozygosity (linear regression p = 0.191; *Appendix 1—figure 1*), indicating that ADO is unlikely to be having a pervasive effect on our data or findings.

**A**

**B**

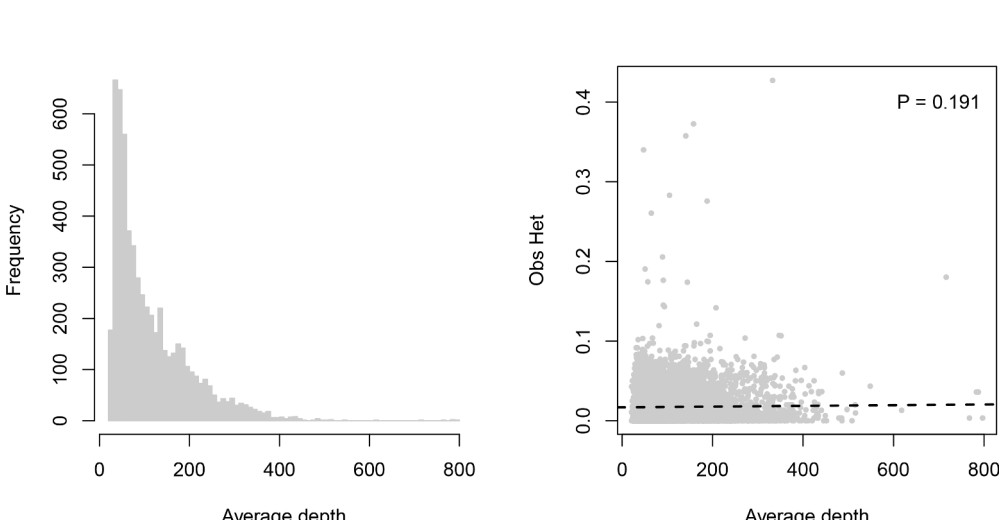

**Appendix 1—figure 1.** Sequencing depth (average number of reads per individual) across loci. Panel (**A**) histogram of depth values. Panel (**B**) relationship between sequencing depth and observed heterozygosity.

To investigate any possible effects of mapping bias, we compared how data presence/absence, numbers of recovered RAD loci, and total number of mapped reads varied between species/populations. Data missingness (the fraction of individuals at a locus without data) never exceeded 20%. We found that CHS/GAL individuals had a higher proportion of sites with 0% missing data than did PIM, although PIM overall never exceeded 10% missing data whereas CHS/GAL did for some markers (*Appendix 1—figure 2*). Regardless, both observations confirmed that the final marker set has low levels of missing data, which should minimize any mapping bias. We also directly examined whether read mapping was biased toward any particular species or population. We did find variation by population (but not by species) in the total number of reads mapped with BWA (*Appendix 1—figure 3*, right panel), but since each population had higher than 95% of reads retained in Stacks (*Appendix 1—figure 3*, center panel), this variation in total reads mapped actually reflects variation among populations in total sequencing output. As a result, there is little variation among populations in the total number of assembled RAD loci (*Appendix 1—figure 3*, left panel), and no consistent difference between species.

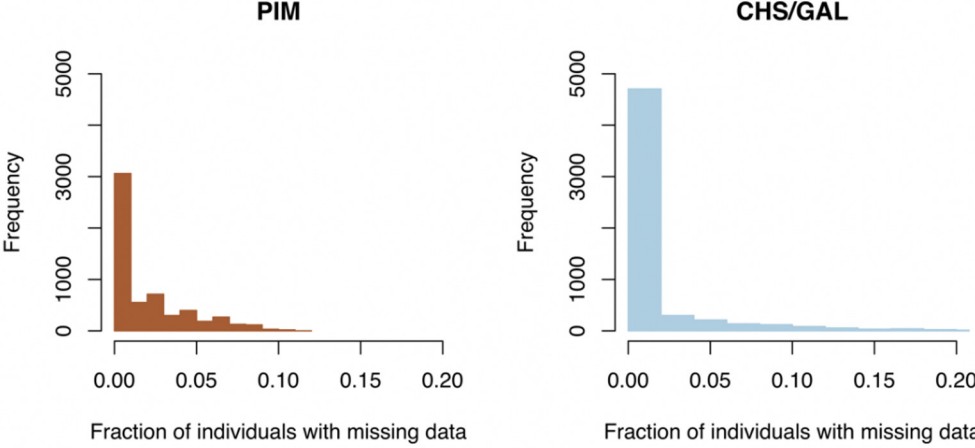

*Appendix 1—figure 2 continued*

**Appendix 1—figure 2.** Clade-specific distributions of missing data fractions across loci. Left panel: histogram of missing data fractions for all PIM populations. Right panel: histogram of missing data fractions for all endemic (CHS/GAL) populations.

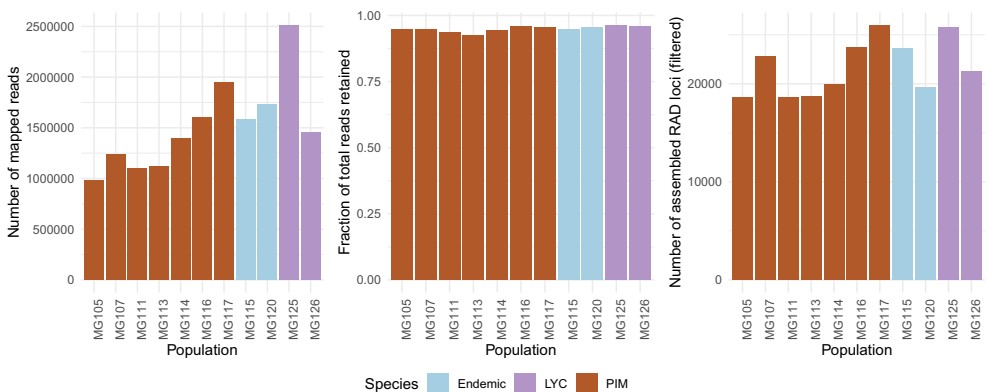

**Appendix 1—figure 3.** Population-specific estimates of the total number of assembled RAD loci (left panel), fractions of total reads retained after assembly (center panel), and total number of reads mapped with BWA (right panel). The total number of mapped reads was on average lower in PIM, yet the average fractions of mapped reads assembled into loci (or 'stacks') and the total number of final RAD loci were roughly equal across populations and species. The differences in numbers of mapped reads point to substantial variation in total sequencing output rather than an inability to map in PIM.

## S7. Local ancestry assignment with HMM

Our HMM was implemented in R and Python (code available at http://github.com/gibsonmatt/gal-tom). We used binned pairwise sequence divergence in 100 kb nonoverlapping windows to define emission probabilities and defined three hidden states (homozygous CHS ancestry, homozygous PIM ancestry, and heterozygous CHS/PIM). We use our HMM to identify regions of recent coalescence between each population.

For a focal individual and in each window, we calculated emission probabilities using three binomial models:

$$P_{CHS} = \binom{d_1}{s} \boldsymbol{\pi}^s (1-\boldsymbol{\pi})^{d_1-s}$$

$$P_{PIM} = \binom{d_2}{s} d_{XY}^s (1-d_{XY})^{d_2-s}$$

$$P_{HET} = \binom{d_3}{s} m^s (1-m)^{d_3-s}$$

where s is the number of sites in the window and $d_1$, $d_2$, and $d_3$ are the median number of differences to MG114, MG115, and the mean of $d_1$ and $d_2$, respectively. Transition probabilities (for the CHS into PIM model) were defined as follows:

$$P[CHS - CHS] = ([1-r] + r) + a$$

$$P[PIM - PIM] = ([1-r] + r) + (1-a)$$

$$P[CHS - PIM] = P[PIM - CHS] = P[HET - CHS] = P[HET - PIM] = r \times a$$

where r and a can be interpreted as proportional to the per-window recombination rate and admixture proportion, respectively. We scaled r by a factor t, which can be interpreted as proportional to the time since admixture. Increasing t will cause the HMM to be more likely to switch between states since recombination will have had more time to break up introgressed blocks. We found that our HMM is relatively insensitive to chosen transition probabilities. For all populations, we chose to use 0.002, 0.48, and 10 for r, a, and t, respectively, as these produced consistent annotations that agreed with patterns of observed nucleotide diversity. Each focal individual in a population was analyzed separately against each of the individuals in the potential donor population. For example, to analyze introgression from CHS → PIM, divergence to all MG114 and MG115 individuals is calculated and used to define emissions.

