## [Decision Letter]

**Acceptance summary:**

This paper reconstructs the history of native and invasive tomatoes in the Galápagos Islands-including species that were first collected by Darwin himself. This is a careful and thoughtful study that describes a highly interesting case of phenotypic convergence for fruit color, driven by the exchange of carotenoid loci between endemic and invasive populations. The work provides a beautiful example of natural experiments that advance our understanding of evolution.

**Decision letter after peer review:**

Thank you for submitting your article "Reconstructing the history and biological consequences of a plant invasion on the Galápagos islands" for consideration by *eLife*. Your article has been reviewed by 2 peer reviewers, and the evaluation has been overseen by a Senior Editor and a Reviewing Editor. The following individual involved in review of your submission has agreed to reveal their identity: Gregory Owens (Reviewer #2).

The reviewers have discussed the reviews with one another and the Reviewing Editor has drafted this decision to help you prepare a revised submission.

Summary:

Tomatoes of the Galapagos Islands are a fascinating system for studying the introduction of alien species as well as their hybridization and competition with native congeners. The current work is based on genotype information from native and introduced tomatoes on the islands as well as mainland individuals. The authors infer multiple introductions using demographic modelling and admixture/introgression between introduced and native populations, nicely highlighting that introduction histories are rarely simple.

Essential revisions:

A major concern are the potential biases inherent in the use of RAD-seq data in combination with a single reference genome. If the authors want to include the dadi analysis (which is based on the joint SFS), they must not only clearly discuss the caveats, but additional analyses must be conducted with other models. Currently, only a very limited set of models is used, with a focus on the median results in the main text – with very large CIs. One would expect the optima (reported in the text) as well as the bootstrap medians to be similar. That they are not suggests a poor fit between the data and the model. As the authors are very well aware, in these types of analyses there is always a 'best fit model' but the 'best' model is not particularly informative if it does not fit the data well."

I am including the full reviews, which provide several suggestions for how to improve the analyses / moderate the claims.

*Reviewer #1:*

Gibson and colleagues use RAD-seq genotyping data from continental and island populations of closely related Solanum species to examine population history and evidence for admixture in the Galapagos islands. This paper combines results from those previously published (Gibson et al., 2020, Molecular Ecology, Gibson et al., 2020, Evolutionary Ecology) and adds in new genotyping and population genetic analyses. The conclusions here are that (i) the wild tomato species *S. pimpinellifolium* was introduced to the Galapagos in the recent past and has caused a decline in the endemic S. cheesmaniae species and S. galapagense species (which appear not to actually be separate species based on the results here) and that (ii) 'borrowed' alleles from the S. cheesmaniae (CHS) and S. galapagense (GAL) have benefitted *S. pimpinellifolium* in some way, enhancing its reproductive success.

The results are broadly consistent with the previous findings published earlier this year both in terms of overall population structure (Gibson et al., 2020, Molecular Ecology) and the introgression of the orange fruit locus (Gibson et al., 2020, Evolutionary Ecology) and represent a clear next-step in this series of papers.

For the population genetic analyses presented, I am concerned about the potential biases introduced from RAD-seq data and in particular the strong conclusions and especially the specific time estimates regarding the migration of the *S. pimpinellifolium* populations. Inherent biases from this type of data and their impacts on basic population genetic parameters have been well-characterized. These were described in Gautier et al., 2013, Arnold et al., 2013, and more recently detailed in Cariou et al., BMC 2016. Arnold et al., 2013 (Mol Ecol) conducted analyses based on simulations and empirical data and found severe biases in genealogical inferences as well as population genetic summary statistics (pi, ThetaW, Tajima's D, FST). Similarly, Gautier et al., (Mol Ecol 2013) showed that allelic dropout in RAD-seq studies biases the inference of genetic variation within and between populations, which was further detailed by Cariou et al., BMC 2016.

Compounded with the issues known for RAD-seq, a single Solanum species is used for alignment of reads, which likely results in a further bias toward apparent lower variation in the diverged island species.

Here, the authors calculate the summary statistics that were already shown to be biased when calculated from RAD-seq data and also go further, using the joint site frequency spectrum for inference via analysis with dadi. Tajima's D is of course a summary statistic based on aspects of the SFS and if it is shown to be biased, I would expect that the numbers of variants assigned to the bins of the JSFS (used in dadi) are problematic.

Even in the best case (i.e., full sequence data), with dadi it is easily possible to find multiple very different demographic models that fit the data equally well (for example earlier migration of fewer individuals vs. more recent migration of a larger number of individuals). Also, in this case, the CIs from the dadi analysis are extremely large, which suggests that the models examined are not very close to reality (or that there are deeper issues with bias due to the nature of the genotype data). For example, in PIM, the bootstrap median estimates for the bottleneck and recovery times are 847 generations ago and 840 generations ago, with confidence intervals of 22-13,591 generations ago for the time to recovery and 0-8000 generations ago for the time to the bottleneck. For the species considered to be island endemics, some estimates were provided but no confidence intervals from that dadi analysis were reported. Even if we are to trust there is no major bias in the RAD-seq genotyping, the dadi do not exlude an ancient natural migration of the *S. pimpinellifolium* species and even seem to support this possibility.

There seems to be pretty strong evidence presented that the orange locus in PIM is due to introgression from CHS.

Gibson and colleagues use RAD-seq genotyping data from continental and island populations of closely related Solanum species to examine population history and evidence for admixture in the Galapagos islands. This paper combines results from those previously published (Gibson et al., 2020, Molecular Ecology, Gibson et al., 2020, Evolutionary Ecology) and adds in new genotyping and population genetic analyses. The conclusions here are that (i) the wild tomato species *S. pimpinellifolium* was introduced to the Galapagos in the recent past and has caused a decline in the endemic S. cheesmaniae species and S. galapagense species (which appear not to actually be separate species based on the results here) and that (ii) 'borrowed' alleles from the S. cheesmaniae (CHS) and S. galapagense (GAL) have benefitted *S. pimpinellifolium* in some way, enhancing its reproductive success.

The results are broadly consistent with the previous findings published earlier this year both in terms of overall population structure (Gibson et al., 2020, Molecular Ecology) and the introgression of the orange fruit locus (Gibson et al., 2020, Evolutionary Ecology) and represent a clear next-step in this series of papers.

For the population genetic analyses presented, I am concerned about the potential biases introduced from RAD-seq data and in particular the strong conclusions and especially the specific time estimates regarding the migration of the *S. pimpinellifolium* populations. Inherent biases from this type of data and their impacts on basic population genetic parameters have been well-characterized. These were described in Gautier et al., 2013, Arnold et al., 2013, and more recently detailed in Cariou et al., BMC 2016. Arnold et al., 2013 (Mol Ecol) conducted analyses based on simulations and empirical data and found severe biases in genealogical inferences as well as population genetic summary statistics (pi, ThetaW, Tajima's D, FST). Similarly, Gautier et al., (Mol Ecol 2013) showed that allelic dropout in RAD-seq studies biases the inference of genetic variation within and between populations, which was further detailed by Cariou et al., BMC 2016.

Compounded with the issues known for RAD-seq, a single Solanum species is used for alignment of reads, which likely results in a further bias toward apparent lower variation in the diverged island species.

Here, the authors calculate the summary statistics that were already shown to be biased when calculated from RAD-seq data and also go further, using the joint site frequency spectrum for inference via analysis with dadi. Tajima's D is of course a summary statistic based on aspects of the SFS and if it is shown to be biased, I would expect that the numbers of variants assigned to the bins of the JSFS (used in dadi) are problematic.

Even in the best case (i.e., full sequence data), with dadi it is easily possible to find multiple very different demographic models that fit the data equally well (for example earlier migration of fewer individuals vs. more recent migration of a larger number of individuals). Also, in this case, the CIs from the dadi analysis are extremely large, which suggests that the models examined are not very close to reality (or that there are deeper issues with bias due to the nature of the genotype data). For example, in PIM, the bootstrap median estimates for the bottleneck and recovery times are 847 generations ago and 840 generations ago, with confidence intervals of 22-13,591 generations ago for the time to recovery and 0-8000 generations ago for the time to the bottleneck. For the species considered to be island endemics, some estimates were provided but no confidence intervals from that dadi analysis were reported. Even if we are to trust there is no major bias in the RAD-seq genotyping, the dadi do not exlude an ancient natural migration of the *S. pimpinellifolium* species and even seem to support this possibility.

There seems to be pretty strong evidence presented that the orange locus in *S. pimpinellifolium* is due to introgression from CHS.

*Reviewer #2:*

This paper reconstructs the invasion history of wild tomatoes in the Galapagos. The authors genotype multiple samples from four species, including native and invasive tomatoes, and their putative mainland progenitors. They find support for multiple Ecuadorian origins of the invasion using demographic modelling. They also find that there has been introgression in multiple instances between native and invasive species and that a fruit color polymorphism in PIM is likely due to introgression from CHS at fruit color loci.

I really enjoyed this paper. It does a very nice and clean job of testing its hypotheses using modern techniques, like dadi or Locator, but also backing them up with basic pop-gen statistics, e.g., π and dxy. The figures are well-made and thoughtful, and the methods are very well documented. I think the results are worth publishing in *eLife*, in particular the finding that fruit color has introgressed is an interesting story that will catch people's interests. With this in mind, I wholeheartedly endorse the paper.

---

## [Author Response]

Essential revisions:A major concern are the potential biases inherent in the use of RAD-seq data in combination with a single reference genome. If the authors want to include the dadi analysis (which is based on the joint SFS), they must not only clearly discuss the caveats, but additional analyses must be conducted with other models. Currently, only a very limited set of models is used, with a focus on the median results in the main text -- with very large CIs. One would expect the optima (reported in the text) as well as the bootstrap medians to be similar. That they are not suggests a poor fit between the data and the model. As the authors are very well aware, in these types of analyses there is always a 'best fit model' but the 'best' model is not particularly informative if it does not fit the data well."I am including the full reviews, which provide several suggestions for how to improve the analyses / moderate the claims.

Thank you for handing our manuscript and for the positive feedback. Below we have addressed each reviewer comment point-by-point, including detailing the text changes as well as additional analyses and figures that have been integrated in the revised submission. Broadly, these revised changes include (i) a thorough quantitative assessment of the impact of allele drop out (ADO) in our RAD seq data; (ii) a reassessment of sequence metadata to evaluate any evidence for mapping bias introduced by our use of a single reference; and (iii) correction of several errors in the dadi inference pipeline that led to exaggerated CIs. We summarize these changes below:

(i) RADseq bias: We agree that the previous manuscript could have better addressed the potential bias in RADseq data caused by allele drop out (ADO). In the revised submission, we now show evidence that genotyping errors caused by polymorphisms in restriction sites are likely not having systematic effects on our results (as detailed further in responses to reviewers below). Briefly, first we note that for species with genome-wide diversity estimates (i.e., π) below 2% (which includes our study species), previous studies (Cariou et al. 2016) indicate that the bias introduced by ADO is negligible. Second, we find no relationship between diversity metrics and sequencing depth (Appendix 1—figure 1); this would be expected if sites sequenced to lower depths experienced drop-out in our dataset. We have amended our methods section to highlight the lack of evidence for ADO (lines 598-602), added an additional supplementary text section (S6), and added Appendix 1—figures 1-3.

(ii) Mapping bias: We agree that the choice of reference genome can have a large influence on diversity metrics when studied species are distantly related. However the four species included in this manuscript are < 0.5My diverged, and prior data indicate that GAL/CHS and PIM are approximately equally related to LYC (the reference species; Pease et al., 2016). Our removal of sites with greater than 20% missing data should also reduce any potential effects of mapping bias if they exist. Nonetheless, to show this more directly, we now investigate data presence/absence, numbers of recovered RAD loci, and total number of mapped reads between species/populations, each of which indicates that mapping to the domesticated tomato reference genome does not generate any consistent or substantial bias between the three wild species. We have added additional text to our methods section to highlight the absence of mapping bias (lines 598-602), provided additional supplementary figures (Appendix 1—figures 2 and 3), and a supplementary text section (S6).

(iii) Demographic models: The largest concern you bring up regards inferences from our demographic models. We agree that the large reported CIs, and the disagreement between optimized estimates and bootstrapped medians, was an unexpected element of our previous analysis, given the natural history and collection data that indirectly support our two main inferences from this analysis (recent, human-mediated introduction of PIM, and older/pre-human introduction/evolution of endemic species). In revisiting our models in response to these concerns, we found three coding errors (described in detail below) that appear to have been responsible for the unexpected difference between optimization and bootstrap runs as well as inflated variance in bootstrapped sampling distributions. After correcting these errors, our revised analyses of the timing and strength of the bottleneck in PIM now have CIs that are more narrow (see revised Table 2), and estimates from optimization and bootstrapping that are in close agreement. For MG115 (CHS), our reanalysis with corrected code indicates that CHS most likely experienced a large expansion between 1114 and 1845 generations ago (rather than a bottleneck, which was a local minimum in our optimization runs), coupled with a very recent and strong contraction. Compared to our estimate of the timing of the bottleneck in PIM, the expansion in CHS is nearly 8 times older. These corrections address the major concerns about our prior poor model fit, while still using only single population SFSs for each species, but also acknowledge that the true history of these populations is likely much more complicated. More complex models which leverage the joint SFS might be possible, but we believe these would be less appropriate for these particular populations/species. We previously constructed a joint SFS from MG114 (PIM) and MG115 (CHS) but found that most variation is actually private and not shared between the two populations (species). We believe this is a realistic consequence of inbreeding coupled with the occurrence of a strong/recent bottleneck in PIM, and indicates that models built around single population SFSs for each species are a preferable alternative to more complex model inferences, that likely go beyond the inferential power of our dataset. Instead, in the revisions we now also provide model-free support for the inferred histories in the form of bootstrapped estimates for genome-wide Tajima’s D (*Table 2—figure supplement 1*). These also clearly support our inference of a recent bottleneck in PIM. We have amended our Results section to reflect the updates to our demographic model fits as well as to include the updated estimates and error intervals for Tajima’s D (lines 220-223).

Reviewer #1:Gibson and colleagues use RAD-seq genotyping data from continental and island populations of closely related Solanum species to examine population history and evidence for admixture in the Galapagos islands. This paper combines results from those previously published (Gibson et al., 2020, Molecular Ecology, Gibson et al., 2020, Evolutionary Ecology) and adds in new genotyping and population genetic analyses. The conclusions here are that (i) the wild tomato species S. pimpinellifolium was introduced to the Galapagos in the recent past and has caused a decline in the endemic S. cheesmaniae species and S. galapagense species (which appear not to actually be separate species based on the results here) and that (ii) 'borrowed' alleles from the S. cheesmaniae (CHS) and S. galapagense (GAL) have benefitted S. pimpinellifolium in some way, enhancing its reproductive success.

Thank you for providing a thorough review of our manuscript. You are correct that there has been historical disagreement about the taxonomic status of the two Galapagos endemic species GAL and CHS. While the most current taxonomic treatments consider them to be separate species, past classifications have not. The distinction does not influence the major inferences from our study – with regard to invasion and introgression involving *S. pimpinellifolium –* so we reference the most recent and widely accepted classifications presented in Darwin et al. (2003).

We have addressed each of your comments below, which includes additional analyses, figures, and text changes where applicable.

The results are broadly consistent with the previous findings published earlier this year both in terms of overall population structure (Gibson et al., 2020, Molecular Ecology) and the introgression of the orange fruit locus (Gibson et al., 2020, Evolutionary Ecology) and represent a clear next-step in this series of papers.

Thank you for taking the time to become familiar with the previous literature supporting this manuscript. We agree that data on non-invasive/mainland PIM population structure, and field observations of invasive and endemic species on the Galapagos, both help to provide a clear context for posing and answering the central questions in the current study.

For the population genetic analyses presented, I am concerned about the potential biases introduced from RAD-seq data and in particular the strong conclusions and especially the specific time estimates regarding the migration of the S. pimpinellifolium populations. Inherent biases from this type of data and their impacts on basic population genetic parameters have been well-characterized. These were described in Gautier et al., 2013, Arnold et al., 2013, and more recently detailed in Cariou et al., BMC 2016. Arnold et al., 2013 (Mol Ecol) conducted analyses based on simulations and empirical data and found severe biases in genealogical inferences as well as population genetic summary statistics (pi, ThetaW, Tajima's D, FST). Similarly, Gautier et al., (Mol Ecol 2013) showed that allelic dropout in RAD-seq studies biases the inference of genetic variation within and between populations, which was further detailed by Cariou et al., BMC 2016.

Thank you for raising this concern. We agree that the manuscript could have more thoroughly addressed potential concerns about biases that can arise from RADseq data, and we have now modified this in the revised text. To clarify, the main source of bias is the potential for polymorphisms in restriction sites leading to allele drop out (ADO). As shown in the papers you cite, genotyping errors caused by ADO can lead to underestimates of SNP-based estimates of polymorphism and, as a result, can bias summary statistics generated from these data (as well as the SFS, but see response to your additional comments below). While we acknowledge this potential concern, two lines of evidence indicate that this effect does not systematically influence the results we present.

First, since ADO is dependent on the occurrence of polymorphisms in restriction cut sites, the impact of ADO is dependent on overall levels of diversity, and any potential bias should roughly scale with levels of sequence diversity. As a result, when polymorphism is low, the potential for bias is also expected to be low. Consistent with this, in their simulation and in silico digestion experiments, Cariou et al. (2016) show that, for species with genome-wide diversity estimates (i.e., π) below 2%, the bias introduced by ADO is “negligible” (pg 5). The average levels of genetic diversity in our species are substantially (an order of magnitude or more) lower than this threshold. In particular, the three wild species in our analysis have estimated average levels of heterozygosity between 0.03% and 0.04% for genome-wide coding sequences and between 0.0% and 0.01% for ultra-conserved orthologs, using high-depth (~30x or more) transcriptome data (Pease et al., 2016). Our estimates from RADseq data in the current study similarly point to extremely low levels of diversity, both on the continent (PIM heterozygosity = 0.025% – 0.033%) and on the Galapagos (PIM heterozygosity = 0.011% – 0.024%). These estimates are generally consistent with previous estimates using other non-RADseq datasets (Caicedo and Shall, 2004). These low levels of polymorphism indicate that the bias introduced by ADO is likely to be negligible in our species. We also note that ADO does not impact estimates of between population divergence (Cariou et al., 2016)

Second, we have also directly reexamined our data for evidence of a potential ADO bias by evaluating conditions under which we would expect to see this effect to be pronounced, if present (see Appendix 1—figure 1). Because high variability in depth is common in RAD datasets, it precludes any “depth-based” methods aimed at detecting ADO (e.g., the approximate Bayesian computation method presented in Cariou et al., 2016). (Indeed, depth across our filtered SNP loci is variable across the genome (Appendix 1—figure 1A) despite being very high on average (average read depth per sample = 114). To reduce this effect of variable depth, we had already filtered sites with fewer than 8 reads supporting either allele, and those where <80% of samples had data.) Instead, we performed a regression of observed heterozygosity (at polymorphic sites; N = 5767) on average depth per sample (Appendix 1—figure 1B) and observe no significant association (P = 0.191), a finding that is not due to lack of power given our large sample size. If ADO was leading to significant bias across the genome, we would expect to observe an association between estimates of genetic diversity (heterozygosity) and read depth, since sites experiencing drop out will also be sequenced to lower depth. This lack of association indicates little evidence for an effect of ADO, similarly indicating that this potential bias is unlikely to impact our inferences from this data.

To address this potential concern in the revised manuscript itself, we have amended our methods section (lines 598-602) to more directly state the possible caveats of using RADseq data. We also added Appendix 1—figure 1 and a brief section (SI section S6) to our supplementary file.

Compounded with the issues known for RAD-seq, a single Solanum species is used for alignment of reads, which likely results in a further bias toward apparent lower variation in the diverged island species.

We agree that choice of reference genome can potentially be a source of bias when compared species are distantly diverged and/or when the species being mapped are differentially related to the reference species. However the data indicate this is not the case for the tomato species presented here. First, all four species (GAL, CHS, PIM, and LYC) are members of the same sub-clade and are less than 0.5 My diverged. Because of this very recent, rapid divergence, species relationships between the individual taxa show substantial gene tree discordance (due to ILS; Pease et al., 2016). However there is no evidence that GAL/CHS and PIM are strongly differentially related to LYC (the reference species); if anything, the data suggests a polytomy among PIM, GAL/CHS, and LYC. Second, our filtering criteria for retaining data was very strict; we removed any sites with higher than 20% missing data, which should also substantially reduce any potential effects of mapping bias.

Third, to more directly investigate any possible effects of mapping bias, we have now also compared how data presence/absence, numbers of recovered RAD loci, and total number of mapped reads varies between species/populations. Appendix 1—figure 2 indicates that data missingness (the fraction of individuals at a locus without data, indicative of failure to map and/or low quality mapping) never exceeds 20% (as expected from our filtering scheme). In these data, we find that CHS/GAL individuals had a higher proportion of sites with 0% missing data than did PIM, although PIM overall never exceeded 10% missing data whereas CHS/GAL did for some markers (Appendix 1—figure 2). Regardless, both observations confirm that final marker set has low levels of missing data, which should minimize mapping bias. We also directly examined whether read mapping was biased towards any particular species or population (Appendix 1—figure 3). We did find variation by population (but not by species) in the total number of reads mapped with BWA (Appendix 1—figure 3, right panel), but since each population had higher than 95% of reads retained in Stacks (Appendix 1—figure 3, center panel), this variation in total reads mapped actually reflects variation among populations in total sequencing output. As a net result, there is little variation among populations in the total number of assembled RAD loci (Appendix 1—figure 3, left panel), and no consistent difference between species.

From the data in Appendix 1—figures 2 and 3, we conclude that reference-based mapping to the domesticated tomato genome bias is likely not responsible for any substantial mapping bias between our three wild species, and does not explain lower variation detected in the endemic species. Instead, the latter is more consistent with the known high selfing rates in these taxa, as well as their historical endemism.

Here, the authors calculate the summary statistics that were already shown to be biased when calculated from RAD-seq data and also go further, using the joint site frequency spectrum for inference via analysis with dadi. Tajima's D is of course a summary statistic based on aspects of the SFS and if it is shown to be biased, I would expect that the numbers of variants assigned to the bins of the JSFS (used in dadi) are problematic.

We jointly address this comment in our response to the related comment below.

Even in the best case (i.e., full sequence data), with dadi it is easily possible to find multiple very different demographic models that fit the data equally well (for example earlier migration of fewer individuals vs. more recent migration of a larger number of individuals). Also, in this case, the CIs from the dadi analysis are extremely large, which suggests that the models examined are not very close to reality (or that there are deeper issues with bias due to the nature of the genotype data). For example, in PIM, the bootstrap median estimates for the bottleneck and recovery times are 847 generations ago and 840 generations ago, with confidence intervals of 22-13,591 generations ago for the time to recovery and 0-8000 generations ago for the time to the bottleneck. For the species considered to be island endemics, some estimates were provided but no confidence intervals from that dadi analysis were reported. Even if we are to trust there is no major bias in the RAD-seq genotyping, the dadi do not exlude an ancient natural migration of the S. pimpinellifolium species and even seem to support this possibility.

Thank you for clearly stating your concerns. We agree that demographic models are significantly prone to bias (given that they are idealized representations of much more complicated true histories they are always incorrect to some degree, even with perfect knowledge of the SFS) and confidence intervals are typically large. This is especially true for very recent demographic events. Uncertainty in parameter estimates is expected to increase closer to the present (Li and Durbin, 2011). Our purpose for including them was not to place hard estimates on the timing of introduction; it was instead to show (1) that we observe a clear excess of rare alleles consistent with a bottleneck model, and (2) that the relative timing of the endemic dispersal event (GAL/CHS) is likely older than the invasive dispersal event (PIM).

As stated in our responses above, our data indicate that ADO is not leading to a systematic bias in our estimates of diversity. This should also extend to the SFS. In their simulations, Arnold et al. (2013) show that ADO affects the SFS by generating an artificial deficit of rare alleles and abundance of intermediate frequency alleles. This pattern would work against the detection of past bottlenecks, so it is unlikely that our inference of bottleneck events in PIM are spuriously generated only by ADO. Genotyping errors brought on by variance in allele sequencing depth are likely a bigger concern, as are the effects of introgression. (Introgression can inject low frequency alleles into the destination population, potentially inflating the observed abundances of rare alleles.) We have mitigated the effects of erroneous genotype calls from low depth sites by using strict filtering schemes (e.g., at least 8X coverage; see methods). Similarly, before estimating the SFS for population MG114 (the population used for invasive bottleneck inference), we removed all sites with evidence for introgression as predicted by our HMM.

As you mention, potential biases introduced by genotyping errors/ADO aside, large confidence intervals might also simply be due to non-identifiability in the limited set of models evaluated. We do not disagree with this statement and have spent a substantial amount of time investigating this during our revisions, especially as the previous –comparatively poor – model fit was puzzling, given other natural history and collections data from the endemic and invasive wild species. Upon reinspection and refitting of our models, we uncovered three programming errors that had substantial effects on our parameter estimates (both in the optimization procedure and in bootstrapping). We thank you for raising questions about these results so we were prompted to catch these errors! The errors were:

1) Only a single grid size was used in dadi for approximation of the diffusion system. The dadi manual suggests that at least two grid sizes are used for proper extrapolation and this we mistakenly did not do this previously. Furthermore, our choice of grid size (30) was likely too small. We have corrected this by now using three grid sizes (30, 40, and 60) and this has substantially reduced variance between optimization runs.

2) A for loop error that resulted in improper bootstrapping. Rather than fitting each replicate using the same starting parameter values, our code mistakenly caused each replicate run to use the previous bootstrap’s model fit as its starting values. This resulted in abnormally long runs (since it quickly approached parameter bounds) and heavily inflated levels of variation between replicates.

3) An error in the dadi optimization pipeline in Portik et al. (2017) that mis-reported the optimal parameter values. It appears that there was an indexing issue in the code provided with their manuscript that led to the mismatching of parameters with their likelihoods after fitting. The reported parameters deviated slightly from the true maximum likelihood estimates. We have addressed this by implementing our own similar optimization procedure (see methods, lines 652-657).

We have now refitted all models using the corrected code. We have revised table 2 (showing parameter estimates and CIs for MG114 – the invasive PIM population)and figure S6 showing the distributions of bootstrapped estimates. During the revisions, we also compared two optimization procedures (BFGS and Nelder-Mead) to ensure that choice of algorithm was not affecting our estimates. In both cases, optimized estimates matched closely with the bootstrapped median estimates (Figure S6) and CIs for TB and TF were smaller by 15-fold or more at the upper bound compared to the previous draft. The estimate for F remained approximately unchanged.

We similarly refitted the same two-epoch model to MG115 (CHS – the endemic species) using the corrected code and performed bootstrapping (Table 2—figure supplement 2). This reanalysis suggested that CHS likely experienced a large expansion between 1114 and 1845 generations ago, coupled with a very recent and strong contraction. The extreme recency of this inferred contraction led us to evaluate additional models as well. However, optimized estimates for single epoch (one size change) as well as a three-epoch (three sizes changes) models were less likely than the two-epoch fit, and so we continue to report the two-epoch model results.

Compared to our estimate of the timing of the bottleneck in PIM, the expansion in CHS is nearly 8 times older, consistent with our understanding of the comparative introduction history of these two species based on historical and contemporary collections and natural history. We have amended our Results section to reflect the change in maximum likelihood estimates for CHS and their associated implications (lines 234-232).

We believe that these corrections address concerns of non-identifiability, especially as it appears that the large CIs were partially a technical error that arose from our prior implementation of these models. We nonetheless acknowledge that the simple models we fit are more or less rough approximations. Unfortunately, fitting more complex models (i.e., to include an ancient speciation event (between CHS and PIM), followed by separate bottlenecks + migration) would require 5-6 additional parameters, as well as estimation of the joint SFS for these two species. We have previously constructed a joint SFS but found that most variation is actually private and not shared between the two populations, suggesting that little additional information is actually contained in the 2D spectrum. We think this is a realistic consequence of inbreeding coupled with the occurrence of strong/recent bottlenecks; thus models built around single population SFSs for each species are a preferable alternative to more complex model inferences that would rely on the joint SFS. We agree with the reviewer that we do not want to go beyond the limits of our data by using it to fit models for which we have insufficient power and resolution, such as these much more complex models.

Instead, as secondary support for our broad conclusions regarding the timing and occurrence of bottlenecks, we now provide additional results (Tajima’s D) which are model-free and do not assume any particular population history. We previously provided a Tajima’s D estimate for MG114 (invasive PIM), and we now provide the same estimate for MG115 (endemic CHS) as well as bootstrapped error intervals (Table 2—figure supplement 1). In accordance with our inference of a more recent bottleneck in PIM, the median value of Tajima’s D (estimated genome-wide) in PIM (-0.49 +/- 0.24) was lower than that of CHS (-0.18 +/- 0.16) indicating a larger excess of rare alleles in PIM consistent with expansion after a recent bottleneck. We have updated text in the Results section to reflect this (lines 220-223 and 240-242).